# LEARNING WITH PRESERVING FOR CONTINUAL MULTITASK LEARNING

## ABSTRACT

Artificial Intelligence (AI) drives advancements across fields, enabling capabilities previously unattainable. Modern intelligent systems integrate increasingly specialized tasks, such as improving tumor classification with tissue recognition or advancing driving assistance with lane detection. Typically, new tasks are addressed by training single-task models or re-training multitask models, which becomes impractical when prior data is unavailable or new data is limited. This paper introduces Continual Multitask Learning (CMTL), a novel problem category critical for future intelligent systems yet overlooked in current research. CMTL presents unique challenges beyond the scope of traditional Continual Learning (CL) and Multitask Learning (MTL). To address these challenges, we propose Learning with Preserving (LwP), a novel approach for CMTL that retains previously learned knowledge while supporting diverse tasks. LwP employs a Dynamically Weighted Distance Preservation loss function to maintain representation integrity, enabling learning across tasks without a replay buffer. We extensively evaluate LwP on three benchmark datasets across two modalities—inertial measurement units of multivariate time series data for quality of exercises assessment and image datasets. Results demonstrate that LwP outperforms existing continual learning baselines, effectively mitigates catastrophic forgetting, and highlights its robustness and generalizability in CMTL scenarios.

## 1 INTRODUCTION

Artificial intelligence is driving progress across numerous critical fields, enabling innovations that were once beyond reach. Increasingly, more specialized and detailed tasks are being integrated into existing intelligent systems, enhancing their capabilities. For example, in medical imaging, tumor classification may evolve to include incremental annotation of tumor shape recognition and tissue density analysis Kaustaban et al. (2022); Freeman et al. (2021). Similarly, intelligent driving assistance systems advance from basic object detection to identifying lanes and recognizing traffic signs Shaheen et al. (2022). Traditionally, additional tasks are integrated by training new single-task models or retraining multitask models, which fall short when previous data is inaccessible or new data is limited. Compiling comprehensive datasets with all labels simultaneously is often unfeasible due to data privacy, resource constraints, or the sequential nature of data collection and annotation. Therefore, labels arrive sequentially, requiring a suitable learning paradigm. Recent works in continual and multitask learning, such as Mirzadeh et al. (2020) and Liao et al. (2022), [1] have addressed these challenges by integrating aspects of continual and multitask learning. However, these approaches often assume access to all tasks or do not generalize new tasks with previous ones. In this paper, we propose *Continual Multitask Learning* (CMTL), a new problem category where input originates from same dataset across tasks, but each task introduces distinct data to label spaces. This reflects real-world scenarios where data drawn from a specific domain are annotated with different attributes over time, requiring models to generate inferences for all the learned attributes for each input. CMTL poses additional challenges compared to traditional Continual Learning (CL) and Multitask Learning (MTL). It requires models to retain knowledge from previous tasks (a CL challenge) and develop shared representations beneficial to multiple tasks (an MTL goal), all while handling new tasks sequentially without access to previous data. In traditional CL, especially task-incremental learning,

---

[1]Further discussion in Appendix A

Figure 1: Comparison among CL, MTL, and CMTL. Two key differences of CMTL compared to the other two scenarios are: (1) inputs stem from a consistent underlying distribution, with labels representing features that any input might have, much like in MTL, and (2) labels are provided sequentially, akin to CL. Models must generalize shared representations while minimizing catastrophic forgetting.

models handle a single task where new classes or labels are introduced over time—like recognizing additional colors in image classification—within the same domain. In contrast, CMTL involves learning different tasks sequentially (e.g., color, shape, size), requiring models to adapt to new task domains while preserving shared representations, as shown in Figure 1. Unlike CL, which focuses on learning new tasks and mitigating catastrophic forgetting Wang et al. (2023; 2022), and MTL, which learns multiple tasks simultaneously, CMTL balances both challenges in a sequential framework. This introduces complexities such as task interference and the need for models to generalize across tasks not available concurrently.

Although CMTL can be classified as a subcategory of Task-incremental Learning (Task-IL) Van De Ven et al. (2022), conventional CL approaches often fail to surpass the performance of multitask models or even simple single-task models under these conditions De Lange et al. (2021); Yoon et al. (2019). Our experiments corroborate this, as shown in Table 1 in Section 4. We hypothesize this shortfall arises because traditional CL methods treat new tasks in isolation, focusing narrowly on task-specific distinctions without considering the broader feature space.

To address these challenges, we bridge the gap by introducing *Learning with Preserving* (LwP). In this novel approach, we preserve previously learned knowledge in a way that remains applicable and beneficial across diverse tasks that may share underlying knowledge structures. This enhancement is designed to maintain both implicitly and explicitly acquired knowledge, ensuring that the learned representations are rich and generalizable enough to facilitate learning in future tasks without interference. The main **contributions** of this paper can be summarized as follows: **a)** We propose a new scenario of continual learning, *CMTL*, highlighting its unique challenges and significance in practical applications where labels arrive sequentially and comprehensive datasets are impractical. **b)** We introduce *Learning with Preserving* (LwP), a novel framework along with a preserving loss function that maintains and distills the integrity of the latent space, ensuring it is conducive to learning across prior and future tasks without relying on a replay buffer. **c)** We demonstrate, through extensive evaluation across two modalities — IMU sensing data (assessing the quality of exercise) and image datasets — that our method outperforms existing baselines, including traditional CL and MTL models, and showcases capabilities in CMTL scenarios.

## 2  PROBLEM FORMULATION: CONTINUAL MULTITASK LEARNING

We propose a subcategory of incremental learning scenarios that closely resemble multitask learning settings. We consider a sequential learning scenario involving $T$ tasks $\{\mathcal{T}_t\}_{t=1}^T$. Each task $\mathcal{T}_t$ is associated with a label space $\mathcal{Y}_t$ and involves learning a mapping $f_t : \mathcal{X} \to \mathcal{Y}_t$. The input space $\mathcal{X}$ is common across all tasks, with input data $\boldsymbol{x} \in \mathcal{X}$ drawn from an identical distribution $P_X$. At each time step $t$, we receive a dataset $D_t = \{(\boldsymbol{x}_i, y_i^t)\}_{i=1}^{n_t}$ sampled from distribution $\mathcal{D}_t$, where $\boldsymbol{x}_i \sim P_X$ is an input sample, $y_i^t \in \mathcal{Y}_t$ is the corresponding label for task $\mathcal{T}_t$. Note that for time $t$, only label $y_i^t$ is available. Other labels $y_i^j$ for $j \neq t$ cannot be observed at time $t$.

Our goal is to find a predictor $\varphi(\boldsymbol{x}; \theta_s, \theta_t) : \mathcal{X} \to \mathcal{Y}_1 \times \mathcal{Y}_2 \times \cdots \times \mathcal{Y}_T$ parameterized by a set of shared parameters $\theta_s$ and task-specific parameters $\theta_t$, such that

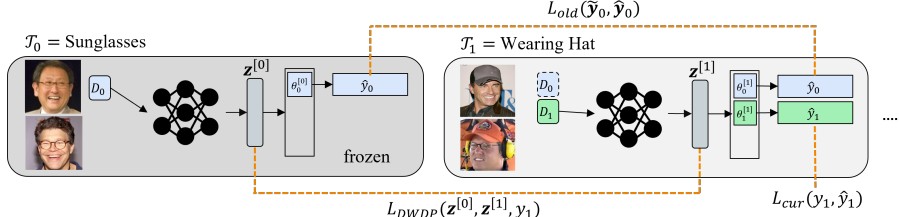

Figure 2: Overview of the LwP training framework on a human face dataset. While learning task $\mathcal{T}1$ (wearing hat) with $\mathcal{L}_{cur}$, the model preserves prior knowledge (sunglasses) through supervised pseudolabeling $\mathcal{L}_{old}$ and implicit knowledge retention via $\mathcal{L}_{DWDP}$.

$$\mathcal{L}(\theta_s, \{\theta_t\}_{t=1}^T) := \sum_{t=1}^T \mathbb{E}_{(\boldsymbol{x}, y^t) \leftarrow \mathcal{D}_t} \left[ \ell \left( y^t, \varphi(\boldsymbol{x}, t; \theta_s, \theta_t) \right) \right], \tag{1}$$

is minimized for some loss function $\ell(\cdot, \cdot)$.

## 3 LEARNING WITH PRESERVING

### 3.1 OVERVIEW

We introduce LwP, a versatile framework designed to effectively manage CMTL scenarios, as depicted in Figure 2. This framework incorporates neural network functions $f_{\theta_s}(\boldsymbol{x})$ to create a shared representation $\boldsymbol{z}$, along with $g_{\theta_t}(\boldsymbol{z})$, which represents task-specific layers for task $t$ and utilizes $\boldsymbol{z}$ to generate predictions for the $t^{th}$ task. This requirement is essential for the model to acquire a shared and generalizable representation space in $\boldsymbol{z}$.

When training the current task $t$, we preserve and freeze the previous model to generate pseudolabels for all the previous $t-1$ tasks. The current model, which is a duplicate of the previous one, includes an additional task-specific layer that will take $\boldsymbol{z}$ as input and learn to predict the current task label $y_t$ using an appropriate supervised loss. Concurrently, the outputs for the previous tasks aim to minimize their supervised loss objectives utilizing pseudolabels from the preceding model.

Following this, we present our key novelty and apply the Dynamically Weighted Distance Preservation (DWDP) loss to preserve the knowledge that has been implicitly learned. Overall, the total objective function for the model while learning task $t$ is defined as:

$$\mathcal{L}_{\text{lwp}} = \lambda_{\text{c}} \mathcal{L}_{\text{cur}}(y_t, \hat{y}_t) + \lambda_{\text{o}} \mathcal{L}_{\text{old}}(\tilde{\boldsymbol{y}}_o, \hat{\boldsymbol{y}}_o) + \lambda_{\text{d}} \mathcal{L}_{\text{DWDP}}(\boldsymbol{z}^{[t]}, \boldsymbol{z}^{[t-1]}, y_t) \tag{2}$$

where

$$\mathcal{L}_{\text{DWDP}} = \frac{1}{N^2} \sum_{i=1}^N \sum_{j=1}^N m_{ij} \left( d(\boldsymbol{z}_i^{[t-1]}, \boldsymbol{z}_j^{[t-1]}) - d(\boldsymbol{z}_i^{[t]}, \boldsymbol{z}_j^{[t]}) \right)^2,$$

$$m_{ij} = \begin{cases} 1, & \text{if } y_i^{[t]} = y_j^{[t]}, \\ 0, & \text{otherwise}, \end{cases}$$

$$\boldsymbol{z}_i^{[t]} = f_{\theta_s^{[t]}}(\boldsymbol{x}_i).$$

$d(\boldsymbol{z}_i, \boldsymbol{z}_j) : \mathbb{R}^d \times \mathbb{R}^d \to \mathbb{R}$ represents either a distance or similarity metric, with $\lambda_{\text{c}}$, $\lambda_{\text{o}}$ and $\lambda_{\text{d}}$ as hyperparameters. $y_t$ and $\hat{y}_t$ denote ground truth and model output of the current task label while $\tilde{\boldsymbol{y}}_o$ and $\hat{\boldsymbol{y}}_o$ denoting previous model's outputs (pseudolabels) and current model's outputs for old tasks, respectively. $\mathcal{L}_{\text{cur}}$ and $\mathcal{L}_{\text{old}}$ represent appropriate supervised learning losses for respective tasks, such as cross entropy or mean squared error. Note that the previous model $\theta^{[t-1]}$ is frozen to

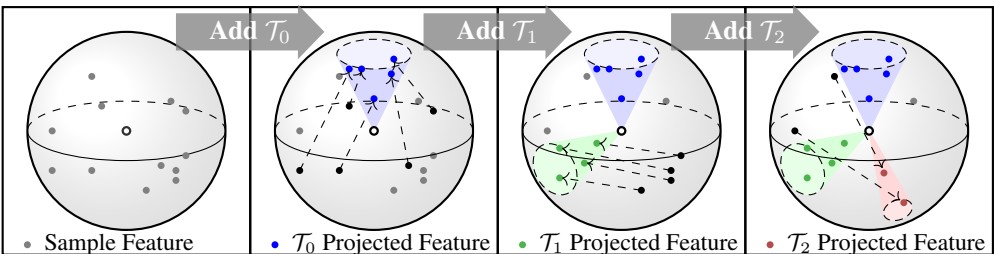

Figure 3: Development of representation space over $\mathcal{T}$. LwP preserves the structure as new tasks are learned.

produce stationary pseudolabels and $\boldsymbol{z}$. In other words, in addition to using pseudolabel to maintain performance on old tasks, we introduce a regularization term aimed at preserving the structure of shared representations by reducing the differences in pairwise similarities (distances) between the model's previous task and the current one if the pairs have the same label for the current task.

## 3.2 PRESERVING IMPLICIT KNOWLEDGE

In the context of CMTL, if $\theta_t$, the task-specific parameters, are simply linear projection layers applied to the final layer of the shared parameters $\theta_s$, we observe that Learning without Forgetting (LwF) Li & Hoiem (2017b) is interpreted as an approximation of the multitask learning objective that encourages the formation of more informative and generalized representation space in $\boldsymbol{z}$.

Motivated by this observation, we show that $\mathcal{L}_{\text{DWDP}}$ is a result of incorporating *implicitly learned knowledge* as an optimization objective. We define such knowledge as the capability of the model's representation to provide an approximate solution to some unknown problem. Therefore, in order to preserve implicitly learned knowledge, we intend to find a loss function that can preserve approximate solutions for *any* problems that can be defined in $\boldsymbol{z}$.

In order to preserve all approximate solutions from the representation space alone, we exploit that kernel methods with the Gaussian kernel are universal approximators Hammer & Gersmann (2003).

Given two sets of representations $Z, Z' \in \mathbb{R}^{n \times d}$, where each row corresponds to $\boldsymbol{z}$, our objective is to ensure that $Z'$ maps to the same Reproducing Kernel Hilbert Space (RKHS) as $Z$ under the Gaussian kernel. To achieve this, we derive a loss function $\mathcal{L}_{pres}$ that encourages the alignment of the pairwise similarities encoded by the Gaussian kernel in both representation spaces.

The Gaussian kernel is defined as $k(\boldsymbol{z}_i, \boldsymbol{z}_j) = \exp\left(-\frac{\|\boldsymbol{z}_i - \boldsymbol{z}_j\|^2}{2\sigma^2}\right)$, where $\boldsymbol{z}_i, \boldsymbol{z}_j \in \mathbb{R}^d$ are representations, and $\sigma > 0$ is the bandwidth parameter controlling the kernel's sensitivity to distance. The Gaussian kernel is a positive definite function, inducing an RKHS $\mathcal{H}$ with an implicit feature mapping $\phi : \mathbb{R}^d \to \mathcal{H}$ such that $k(\boldsymbol{z}_i, \boldsymbol{z}_j) = \langle \phi(\boldsymbol{z}_i), \phi(\boldsymbol{z}_j) \rangle_{\mathcal{H}}$.

For a set of representations $Z$, the Gram matrix $K(Z) \in \mathbb{R}^{n \times n}$ is constructed with entries $K_{ij}(Z) = k(\boldsymbol{z}_i, \boldsymbol{z}_j)$. Similarly, we construct $K(Z')$ for $Z'$. Our goal is to align $K(Z)$ and $K(Z')$ such that the pairwise similarities in $Z'$ match those in $Z$. This alignment ensures that $Z$ and $Z'$ are mapped to the same locations in the RKHS up to an isometry.

To formalize the alignment objective, we define the loss function $\mathcal{L}_{pres}$ as the squared Frobenius norm of the difference between the two kernel matrices:

$$\mathcal{L}_{pres}(Z, Z') = \|K(Z) - K(Z')\|_F^2 = \sum_{i=1}^{n} \sum_{j=1}^{n} \left(k(\boldsymbol{z}_i, \boldsymbol{z}_j) - k(\boldsymbol{z}_i', \boldsymbol{z}_j')\right)^2. \tag{3}$$

Minimizing $\mathcal{L}_{pres}$ with respect to $Z'$ (while keeping $Z$ fixed) encourages the kernel matrices to become identical, i.e., $K(Z') \approx K(Z)$. This implies that for all pairs $(i, j)$, $k(\boldsymbol{z}_i', \boldsymbol{z}_j') \approx k(\boldsymbol{z}_i, \boldsymbol{z}_j)$.

By minimizing $\mathcal{L}_{pres}$, we effectively align the images of $Z$ and $Z'$ under the feature map $\phi$:

$$\langle \phi(\boldsymbol{z}_i), \phi(\boldsymbol{z}_j) \rangle_{\mathcal{H}} \approx \langle \phi(\boldsymbol{z}_i'), \phi(\boldsymbol{z}_j') \rangle_{\mathcal{H}}, \quad \forall i, j. \tag{4}$$

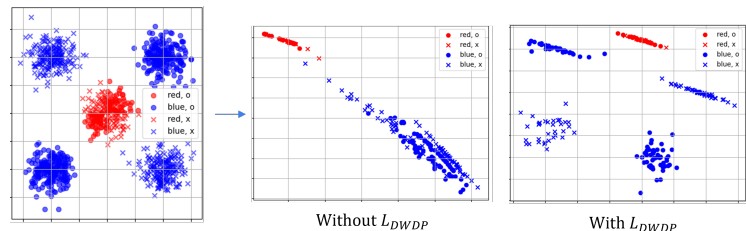

Figure 4: The impact of $\mathcal{L}_{\text{DWDP}}$ on a two-dimensional toy dataset, where $y_1$ (O vs. X) indicates an XOR problem and $y_2$ (blue vs. red) signifies a concentric circle problem. The figure shows the representation space after training on $y_2$ without $\mathcal{L}_{\text{DWDP}}$ (left) and with $\mathcal{L}_{\text{DWDP}}$ (right). The latter successfully preserves the cluster structures of the former representation, which is advantageous for learning $y_1$ in subsequent phases.

This alignment implies that there exists an isometry $T : \mathcal{H} \to \mathcal{H}$ such that:

$$\phi(\boldsymbol{z}_i') = T(\phi(\boldsymbol{z}_i)), \quad \forall i. \tag{5}$$

For any function $f \in \mathcal{H}$, there exists a weight vector $w \in \mathcal{H}$ such that $f(\boldsymbol{z}) = \langle w, \phi(\boldsymbol{z}) \rangle_{\mathcal{H}}$. The evaluation of $f$ at $\boldsymbol{z}_i'$ becomes:

$$f(\boldsymbol{z}_i') = \langle w, \phi(\boldsymbol{z}_i') \rangle_{\mathcal{H}} = \langle w, T(\phi(\boldsymbol{z}_i)) \rangle_{\mathcal{H}}. \tag{6}$$

Because $T$ is an isometry, its adjoint $T^*$ is also an isometry, and we can write:

$$f(\boldsymbol{z}_i') = \langle T^* w, \phi(\boldsymbol{z}_i) \rangle_{\mathcal{H}}. \tag{7}$$

Define $w' = T^* w$ and $f'(\boldsymbol{z}) = \langle w', \phi(\boldsymbol{z}) \rangle_{\mathcal{H}}$. Then:

$$f(\boldsymbol{z}_i') = f'(\boldsymbol{z}_i), \quad \forall i. \tag{8}$$

Thus, $Z'$ becomes an alternative representation that is functionally equivalent to $Z$ in terms of any operations performed within the RKHS induced by the Gaussian kernel. Now, consider a learning problem defined on $Z$:

$$\min_{f \in \mathcal{H}} \frac{1}{n} \sum_{i=1}^{n} \ell(f(\boldsymbol{z}_i), y_i) + \Omega(f), \tag{9}$$

and the corresponding problem on $Z'$:

$$\min_{f \in \mathcal{H}} \frac{1}{n} \sum_{i=1}^{n} \ell(f(\boldsymbol{z}_i'), y_i) + \Omega(f). \tag{10}$$

Using the relationship $f(\boldsymbol{z}_i') = f'(\boldsymbol{z}_i)$, the loss terms satisfy $\ell(f(\boldsymbol{z}_i'), y_i) = \ell(f'(\boldsymbol{z}_i), y_i)$. Since $\|f\|_{\mathcal{H}} = \|f'\|_{\mathcal{H}}$, the regularization terms are equal: $\Omega(f) = \Omega(f')$. Thus, the risk functionals for the problems on $Z$ and $Z'$ are equivalent when considering $f$ and $f'$:

$$\frac{1}{n} \sum_{i=1}^{n} \ell(f(\boldsymbol{z}_i'), y_i) + \Omega(f) = \frac{1}{n} \sum_{i=1}^{n} \ell(f'(\boldsymbol{z}_i), y_i) + \Omega(f'). \tag{11}$$

Because the risk functionals are equivalent, the optimal solutions $f^*$ obtained on $Z'$ correspond to the optimal solutions $f'^*$ on $Z$ via the isometry $T^*$:

$$f^*(z_i') = f'^*(z_i). \tag{12}$$

This means any model trained on $Z$ can be transformed to a model on $Z'$ with identical performance, and vice versa.

Through empirical observation, we have determined that maintaining the squared Euclidean distance instead of $k$ directly leads to enhanced performance. This is likely because the non-exponentiated distance metric more effectively retains the global structure of the representation space. Refer to Appendix 4.6 for the experimental data. Additionally, in Appendix C, we show that the difference in RBF kernel values is bounded by the difference in the squared $L^2$ norm.

Hereby we define a family of such losses that preserve some distance (or similarity) metric between pairs of representations as the following:

$$\mathcal{L}_{\text{pres}}(z, z') = \frac{1}{N^2} \sum_{i=1}^{N} \sum_{j=1}^{N} \left( d(z_i, z_j) - d(z_i', z_j') \right)^2, \tag{13}$$

where $d$ represents either a distance or a similarity function. Note that it no longer needs to be a kernel to include a broader variety of metrics.

### 3.3 Dynamic Weighting

$\mathcal{L}_{\text{pres}}$ is designed to maintain the implicitly learned knowledge of the input data in the representation space. However, in scenarios where there are distinct classes or labels, this loss can conflict with other objectives, such as separating distinct classes.

To address this issue, we introduce the Dynamically Weighted Distance Preservation (DWDP) Loss, $\mathcal{L}_{\text{DWDP}}$. This loss function adapts the preservation loss by applying a dynamic mask $m_{ij}$, which controls the contribution of each pairwise comparison based on their label similarity. The intuition behind this modification is to deactivate the preservation requirement for pairs with different labels, thus preventing conflicts with the separation objectives.

The dynamic mask $m_{ij}$ is defined as follows:

$$m_{ij} = \begin{cases} 1, & \text{if } y_i^{[t]} = y_j^{[t]}, \\ 0, & \text{otherwise,} \end{cases} \tag{14}$$

where $y^{[t]}$ represents the labels of the current task.

Thus, the DWDP Loss is then given by:

$$\mathcal{L}_{\text{DWDP}}(z^{[t-1]}, z^{[t]}, y^{[t]}) = \frac{1}{N^2} \sum_{i=1}^{N} \sum_{j=1}^{N} m_{ij} \left( d(z_i^{[t-1]}, z_j^{[t-1]}) - d(z_i^{[t]}, z_j^{[t]}) \right)^2 \tag{15}$$

Consequently, this modification alleviates the objective conflict issue at the cost of reducing the scope for preservation to intraclass sets of the current task. Our detailed pseudo-code algorithm is presented in Appendix B.

## 4 Evaluation

### 4.1 Overview

We present a set of experiments designed to rigorously validate our approach using three benchmark datasets that span multiple modalities. **1)** We conduct a comprehensive performance evaluation of our method, LwP, comparing it to state-of-the-art CL techniques. This assessment focuses on the

average accuracy across all tasks after training is completed. **2)** We analyze the extent of catastrophic forgetting in each model, employing the Backward Transfer (BWT) metric Lopez-Paz & Ranzato (2017a) to quantify the accuracy degradation over successive tasks. **3)** We benchmark the rate of performance improvement per train iteration not only against other baseline CL models but also against MTL oracles, which serve as an upper bound for the problem. **4)** We explore the effects of dynamic weighting and various distance/similarity functions, $d$, as proposed in other studies, on the performance metrics of LwP.

Additionally, in the Appendix D, we explain more detailed setup including attributes and conduct further evaluations through additional experiments. A detailed comparison with MTL methods is presented in Appendix D.2. Furthermore, we supply supplemental diagrams that illustrate the progression of the accuracy over time for both the PhysiQ and FairFace datasets, which can be found in Appendix D.3. Appendix D.4 elaborates on the influence of the number of training examples on the overall performance of each model. In Appendix D.5, we benchmark the rate of performance improvement per train iteration not only against other baseline CL models but also against MTL methods. In Appendix D.6, we analyze the impact of model size and image size on the performance of all the methods. Additionally, in Appendix D.7, we explore training the first 5 tasks of the CelebA dataset using an MTL scheme, followed by a CL setting for the remaining 5 tasks.

## 4.2 EXPERIMENT SETUP

**Datasets** We utilize three datasets from two distinct modalities, each structured for task-incremental learning. In this setting, each task is only exposed to a subset of training samples:

*The CelebA dataset* Liu et al. (2018), consisting of 200,000 images with 40 facial attributes. For our work, we focus on 10 of the most balanced attributes. The train dataset is equally subdivided for each task, leading to 20,000 images per task. For simplicity, input images are resized to 32x32.

*The PhysiQ dataset* Wang & Ma (2023), which contains approximately 4,500 samples collected using inertial measurement units (IMUs) to capture the quality of physical exercises. The data is collected on accelerometer and gyroscope modality of 50 Hz sampling rate for 31 participants with three attributes. Each task corresponds to one of these attributes with around 1,500 samples.

*The Fairface dataset* Karkkainen & Joo (2021), which includes 100,000 images with three attributes. Following the same subdivison procedure, the dataset results in containing approximately 33,333 images with a resolution of 128x128 per task. Not only the tasks differ from those of CelebA, but also the images are not resized in order to show our approach is scalable.

**Baselines** Our primary emphasis is on CL baselines since integrating many MTL methods into CMTL often requires substantial modifications to accommodate the incremental characteristics of CMTL. For CL, we compare against Online Bias Correction (OBC) Chrysakis & Moens (2023), Dual View Consistency (DVC) Gu et al. (2022), Dark Experience Replay (DER) Buzzega et al. (2020), DER++ Boschini et al. (2022), Function Distance Regularization (FDR) Benjamin et al. (2019), Experience Replay (ER) Robins (1995); Ratcliff (1990), Gradient-based Sample Selection (GSS) Aljundi et al. (2019b), online Elastic Weight Consolidation (oEWC) Kirkpatrick et al. (2017), Synaptic Intelligence (SI) Zenke et al. (2017), and Learning without Forgetting (LwF) Li & Hoiem (2017b). In addition, we compare our approach with MTL methods, which are detailed in Appendix D.2. These include the basic MTL approach of training all tasks simultaneously with different predictors Caruana (1997), as well as more advanced techniques like PCGrad Yu et al. (2020), Impartial MTL (IMTL) Liu et al. (2021), and NashMTL Navon et al. (2022). We also include a single task learning (STL) baseline, where each task is learned separately. For the choice of distance metric $d$, we test common options such as Euclidean distance and cosine similarity, as well as loss functions designed to preserve relational knowledge, such as those proposed in RKD Park et al. (2019) and Co2L Cha et al. (2021).

**Model Architectures** We use an untrained ResNet-18 for CelebA and FairFace datasets. Each task is predicted after a linear projection layer applied to the flattened last shared layer $z \in \mathbb{R}^{512}$. Similarly, for PhysiQ, we use a 3-layer 1DCNN model with $z \in \mathbb{R}^{128}$ final shared layer connected to task-specific linear layers. We evaluate additional architectures and image sizes in Appendix D.6.

## 4.3 COMPREHENSIVE COMPARISON

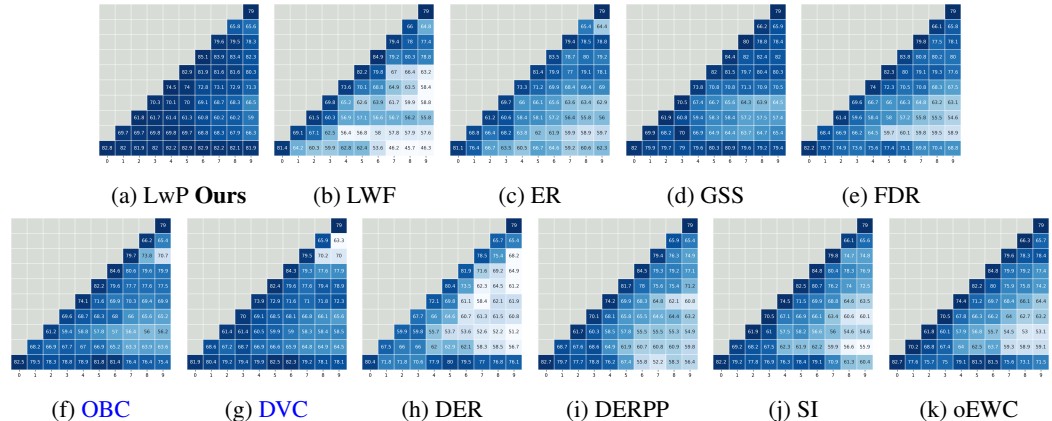

Figure 5: Matrices showcasing the accuracy progression for various models for Dataset CelebA. Each column corresponds to an iteration of the task, arranged sequentially from left to right. We generate the confusion matrices normalized on all the tasks for all the models for consistency.

Table 1: Accuracy Percentage Comparison Across Models and Datasets

| Method Type | Model | CelebA (10 Tasks) | PhysiQ (3 Tasks) | FairFace (3 Tasks) |
|---|---|---|---|---|
| STL | - | $72.230 \pm 7.297$ | $87.167 \pm 10.102$ | $64.435 \pm 3.660$ |
| CL | LwF | $64.626 \pm 10.806$ | $69.952 \pm 21.090$ | $61.034 \pm 6.162$ |
| | oEWC | $69.666 \pm 9.019$ | $82.640 \pm 12.166$ | $63.604 \pm 3.122$ |
| | ER | $67.598 \pm 7.452$ | $76.798 \pm 16.347$ | $63.220 \pm 4.730$ |
| | SI | $68.735 \pm 10.545$ | $83.727 \pm 11.828$ | $63.359 \pm 3.451$ |
| | GSS | $71.680 \pm 8.468$ | $85.741 \pm 10.950$ | $64.230 \pm 3.918$ |
| | FDR | $69.514 \pm 8.917$ | $71.859 \pm 18.687$ | $63.709 \pm 3.151$ |
| | DER | $70.703 \pm 8.388$ | $84.796 \pm 11.168$ | $64.114 \pm 3.484$ |
| | DERPP | $67.693 \pm 9.425$ | $82.838 \pm 13.775$ | $63.806 \pm 3.694$ |
| | DVC | $71.441 \pm 7.640$ | $85.100 \pm 10.381$ | $63.848 \pm 3.193$ |
| | OBC | $70.829 \pm 8.267$ | $83.999 \pm 11.377$ | $63.872 \pm 3.449$ |
| CMTL | **LwP** | $\mathbf{73.484 \pm 8.019}$ | $\mathbf{88.242 \pm 12.010}$ | $\mathbf{66.482 \pm 3.138}$ |

In this experiment, we evaluate the performance of Our LwP against several state-of-the-art CL methods. All methods, except for LwF and LwP, are provided with a buffer size of 512 for the CelebA and FairFace datasets, and 46 for the PhysiQ dataset, corresponding to approximately 2-3% of the training set for each dataset. Each model is trained five times using different random seeds. The standard training protocol consists of 20 epochs, with a batch size of 256 for image-based datasets and 32 for PhysiQ, coupled with early stopping. For PhysiQ, we only compare the average accuracy across the final task iteration due to the training instability caused by smaller dataset size. Table 1 reports the average test accuracy, along with the standard deviation over five runs for each method and dataset. Fig. 5 visualizes the progression of task accuracy in task iterations (left to right). Additional results are provided in Appendix D.3

Table 1 highlights that LwP consistently achieves superior performance across all three benchmarks and is the only method to exceed the Single Task Learning (STL) baseline. This suggests that other continual learning methods likely experience significant task interference. Additional results with MTL are provided in Appendix D. Furthermore, our approach is modality-agnostic, as evidenced by LwP's ability to generalize across different domains. This is demonstrated by the results on the PhysiQ dataset from the IMU sensor domain, which underscores LwP's robustness against challenges unique to non-image-based tasks.

*The results suggest that LwP demonstrates competitive performance compared to existing continual learning methods across a range of benchmarks in CMTL settings. LwP consistently achieves higher*

*accuracy than other approaches, including surpassing the STL baseline, indicating its potential to reduce task interference and catastrophic forgetting in different modalities.*

## 4.4 BACKWARD TRANSFER

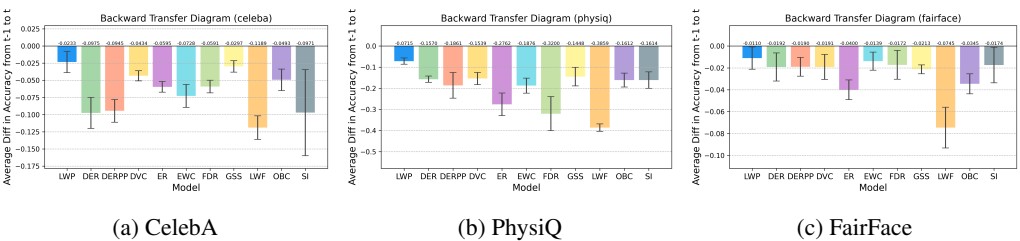

(a) CelebA  (b) PhysiQ  (c) FairFace

Figure 6: Backward Transfer Diagrams for Various Datasets

The Backward Transfer Lopez-Paz & Ranzato (2017a) is a metric to evaluate the influence of learning the current task on the performance of previous tasks. A positive backward transfer value indicates that, on average, accuracies on the previous tasks have increased during the current task iteration and vice versa. It is defined as:

$$BWT = \frac{1}{T-1} \sum_{i=1}^{T-1} R_{T,i} - R_{i,i}, \tag{16}$$

where $T$ is the index of the current task, $i$ is an index of previous tasks ranging from 1 to $T-1$, $R_{T,i}$ is the accuracy on task $i$ after training up to task T, and $R_{i,i}$ is the accuracy on task $i$ after learning.

As illustrated in Fig. 6, we observe that LwP outperforms all baselines in terms of BWT across all benchmarks. This result is consistent with the visualization shown in Fig. 3, where LwP can maintain the accuracy of each task since its initial training.

## 4.5 REPRESENTATION SPACE VISUALIZATION VIA T-SNE

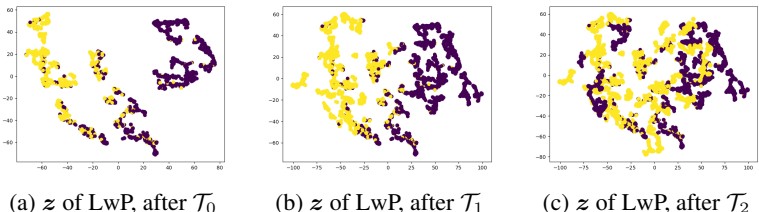

(a) $z$ of LwP, after $\mathcal{T}_0$  (b) $z$ of LwP, after $\mathcal{T}_1$  (c) $z$ of LwP, after $\mathcal{T}_2$

Figure 7: Representation space progression over task iteration. Colors indicate different label values.

Fig. 7 shows the 2D visualization of the representation $z$ for each model trained on the PhysiQ dataset. It was constructed using the dimensionality reduction algorithm t-SNE van der Maaten & Hinton (2008) on the PhysiQ test dataset at the end of each task iteration. x and y axes represent the two new dimensions created by the algorithm to project the high-dimensional data onto a 2D plane. Note these dimensions do not have an intrinsic meaning and rather constructed to reflect the relative distances between data points in the high-dimensional space. As demonstrated, the $z$ produced by LwP maintains coherent cluster formations as it progressively learns new tasks without introducing considerable distortions when compared to the baseline model. This behavior is comparable to the example provided with the toy dataset depicted in Fig. 4.

## 4.6 THE ABLATION STUDY OF EFFECTIVENESS OF THE LOSS FUNCTION

To evaluate the impact of the proposed loss function, we perform experiments by selectively disabling the dynamic weighting feature and comparing it with other loss functions that also aim to

Table 2: Ablation comparison on $\mathcal{L}_{\text{DWDP}}$ implementation

| Method on PhysiQ | LwP ($L^2$) | LwP (Cosine) | LwP (RBF) | IRD (Co2L) | RKD |
|---|---|---|---|---|---|
| Dynamic Weighting | **88.2 ± 12.0** | 85.4 ± 13.1 | 84.5 ± 13.7 | 86.4 ± 11.5 | 85.1 ± 13.3 |
| W/o Dynamic Weighting | 86.0 ± 12.3 | 84.1 ± 14.4 | 84.8 ± 14.5 | 79.9 ± 17.1 | 85.9 ± 11.9 |

preserve structures. In our evaluation, we include CO2L Cha et al. (2021), RKD Park et al. (2019), and two novel variation of baselines: cosine similarity and the RBF kernel as described in eq. 3.

The findings in Table 2 indicate that the loss function with both dynamic weighting and Euclidean distance consistently surpasses the other options. We believe that the effectiveness of Euclidean distance with dynamic weighting is due to its loss not being normalized across batches, unlike previously proposed approaches.

## 5 RELATED WORK

MTL enhances generalization and computational efficiency by leveraging shared representations across related tasks Caruana (1997); Sener & Koltun (2018). However, optimizing multiple objectives often presents conflicting gradients. Approaches like the Multiple Gradient Descent Algorithm (MGDA) Sener & Koltun (2018) seek Pareto optimal solutions through convex combinations of task-specific gradients, while Gradient Surgery (PCGrad) Yu et al. (2020) projects conflicting gradients onto the normal plane of each other to reduce interference. Navon et al. Navon et al. (2022) modeled gradient combination as a cooperative bargaining game to ensure fairness among tasks. Loss balancing is also crucial, with methods like IMTL Liu et al. (2021) incorporating both gradient and loss balancing mechanisms. CL enables sequential task learning without catastrophic forgetting Ratcliff (1990); Robins (1995). Techniques like MER Riemer et al. (2018) focus on maximizing knowledge transfer while minimizing interference, while HAL Chaudhry et al. (2021) anchors past knowledge to prevent representation drift. Bridging MTL and CL, continual multitask learning aims to manage performance across sequential and concurrent tasks Wu et al. (2023), using methods like MC-SGD Mirzadeh et al. (2020) to enhance CL by leveraging linear mode connectivity. Task-free CL Aljundi et al. (2019a) eliminates task boundaries. More detailed discussions are available in the Appendix A.

## 6 CONCLUSION

We explored the limitations of existing continual learning methods in CMTL. Our findings show that conventional approaches often underperform compared to single-task models, largely due to their focus on preserving explicit information while neglecting broadly useful, implicit features. To address this, we introduced *Learning with Preserving* (LwP) with a dynamically weighted distance preservation function. This approach maintains the structure of the representation space, preserving implicit knowledge without needing replay buffers, making it especially valuable in privacy-sensitive domains like healthcare. Our experiments across various datasets demonstrated that LwP surpasses state-of-the-art baselines and outperforms single-task models, consistently retaining accuracy and mitigating catastrophic forgetting. The results emphasize the importance of preserving implicit knowledge and the effectiveness of our loss function. Future work could explore LwP's application with non-stationary dataset or unlabeled data (i.e., investigation on KL divergence vs. LwP performance), and its integration with pre-trained foundation models.

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

# A RELATED WORKS

## A.1 MULTITASK LEARNING

Multitask learning (MTL) has been extensively explored for its ability to leverage shared representations across multiple related tasks, thereby enhancing generalization and computational efficiency Caruana (1997); Sener & Koltun (2018); Javaloy & Valera (2021); Gardner et al. (2022); Kim et al. (2022); Cao et al. (2022); Yang et al. (2024); Liu et al. (2021). In MTL, models are trained on multiple tasks simultaneously, with the assumption that learning tasks together allows the model to capture commonalities and differences among tasks, leading to better performance than training each task separately.

A central challenge in MTL is the optimization of multiple objectives, which often present conflicting gradients that can impede the convergence and performance of the model. To address this, various gradient balancing approaches have been proposed. Sener and Koltun Sener & Koltun (2018) introduced the Multiple Gradient Descent Algorithm (MGDA), which seeks Pareto optimal solutions by finding a convex combination of task-specific gradients. Building on this, Yu et al. Yu et al. (2020) proposed Gradient Surgery (PCGrad), which directly modifies conflicting gradients by projecting them onto the normal plane of each other to reduce negative interference. More recently, Navon et al. Navon et al. (2022) approached MTL from a game-theoretic perspective, modeling the gradient combination step as a cooperative bargaining game and employing the Nash Bargaining Solution to ensure proportional fairness among tasks.

In addition to gradient balancing, loss balancing is crucial for stable and unbiased learning in MTL. Liu et al. Liu et al. (2021) introduced IMTL, which incorporates both gradient and loss balancing mechanisms. Their method, IMTL-G, ensures unbiased updates to task-shared parameters by finding the geometric angle bisector of task gradients, while IMTL-L automatically learns loss weighting parameters to harmonize the scales of different task losses.

While these MTL methods have advanced the ability to learn multiple tasks simultaneously, they typically assume that all task data is available at training time and can be processed jointly. This assumption does not hold in scenarios where tasks and their associated data arrive sequentially, as in our defined problem, Continual Multitask Learning (CMTL). In such cases, models must learn new tasks without access to all previous data, and ideally, they should leverage new tasks to improve performance on prior tasks.

Our work differs from traditional MTL approaches by addressing the sequential arrival of tasks and data, where tasks are learned iteratively rather than simultaneously. Unlike MTL methods that focus on balancing gradients and losses across tasks trained together, our approach must handle the challenge of incorporating new tasks without retraining on previous tasks' data. Furthermore, we introduce mechanisms to utilize new task data to enhance the model's generalizability on earlier tasks, which is not considered in standard MTL frameworks.

## A.2 CONTINUAL LEARNING (CL)

CL aims to enable models to learn sequentially from a stream of tasks without forgetting previously acquired knowledge, addressing the challenge of catastrophic forgetting Ratcliff (1990); Robins (1995). Various methods have been developed to tackle this problem, broadly categorized into *rehearsal-based methods*, *knowledge distillation*, and *regularization-based techniques*.

**Rehearsal-based methods** Early works such as Ratcliff (1990); Robins (1995) introduced Experience Replay (ER), where old data samples are mixed with current ones during training. Building upon this concept, Robins Robins (1995) explored pseudorehearsal techniques. More recent methods like Meta-Experience Replay (MER) Riemer et al. (2018) reformulate ER within a meta-learning framework, aiming to enhance knowledge transfer between past and present tasks while reducing interference. Gradient-based Sample Selection (GSS) Aljundi et al. (2019b) modifies ER by selecting optimal examples for storage in the memory buffer, improving retention of past knowledge. Another method, Hindsight Anchor Learning (HAL) Chaudhry et al. (2021), augments ER with an additional goal to prevent forgetting key data points. Gradient Episodic Memory (GEM) Lopez-Paz & Ranzato (2017b) and its more efficient variant Averaged-GEM (A-GEM) Chaudhry et al. (2018b) use previ-

ous training data to impose optimization constraints on the current update, ensuring better retention of learned information. Additionally, Yoon et al. Yoon et al. (2019) introduced Additive Parameter Decomposition (APD), an architectural approach that represents the parameters for each task as a sum of task-shared and task-adaptive parameters. APD ensures scalability and order-robustness by preventing catastrophic forgetting and addressing order-sensitivity through parameter decomposition. Lastly, Aljundi et al. Aljundi et al. (2019a) introduce task-free continual learning, eliminating the need for task boundaries and enabling more flexible adaptation to new tasks without explicit task identifiers.

These methods, while effective in certain scenarios, rely heavily on storing and replaying data from previous tasks, which may not be feasible due to privacy concerns or memory constraints. In contrast, our approach does not require storing raw data from previous tasks. Instead, we utilize pseudolabels generated by the frozen previous model and introduce a novel regularization term to preserve the structure of shared representations. This enables the model to retain and improve upon prior knowledge without explicit rehearsal.

**Knowledge Distillation**   Methods leveraging Knowledge Distillation Hinton (2015) address the issue of forgetting by using a previous iteration of the model as a teacher. Learning Without Forgetting (LwF) Li & Hoiem (2017a) generates a softened version of the model's current outputs on new data at the onset of each task, minimizing output drift throughout training. iCaRL Rebuffi et al. (2017) combines distillation with replay techniques, using a memory buffer to train a nearest-mean-of-exemplars classifier while applying a self-distillation loss to preserve learned representations across tasks. Moreover, Li et al. Li et al. (2019) proposed a continual learning method tailored for sequence-to-sequence tasks, leveraging compositionality to enable knowledge transfer and prevent catastrophic forgetting. Their approach extends traditional label prediction continual learning methods to handle more complex tasks like machine translation and instruction learning.

While these methods use knowledge distillation to maintain performance on old tasks, they typically focus on preserving output logits or feature representations without considering the underlying relational structure between data points. Our method extends this idea by not only preserving the output predictions via pseudolabels but also maintaining the pairwise relationships in the representation space through our Dynamically Weighted Distance Preservation (DWDP) loss. This helps in better retaining the learned structure and prevents the model from drifting away from previously acquired knowledge.

**Regularization-based techniques**   These methods modify the loss function to include a penalty that restricts changes to the model's parameters. Examples include Elastic Weight Consolidation (EWC) Duncker et al. (2020), its online variant (oEWC) Kirkpatrick et al. (2017), Synaptic Intelligence (SI) Zenke et al. (2017), and Riemannian Walk (RW) Chaudhry et al. (2018a). In contrast, architectural methods such as Progressive Neural Networks (PNN) Rusu et al. (2016) incrementally expand the model by adding new networks for each task, which leads to increased memory usage. To address this, methods like PackNet Mallya & Lazebnik (2018) and Hard Attention to the Task (HAT) Serra et al. (2018) reuse the same architecture for multiple tasks, dynamically allocating resources to prevent performance degradation. Recent advances include a generalized framework with additional loss functions proposed by Wang et al. Wang et al. (2024). Another promising architectural method is task-conditioned hypernetworks Von Oswald et al. (2019), which generate weights for the target network based on task identity. These hypernetworks do not need to recall all input-output relationships for previously seen tasks, as they instead rehearse task-specific weight realizations. Moreover, Adel et al. Adel et al. (2019) introduced Continual Learning with Adaptive Weights (CLAW), which employs a probabilistic modeling approach to adaptively identify which parts of the network should be shared across tasks in a data-driven manner. This method balances between modeling each task separately to prevent catastrophic forgetting and sharing components to allow transfer learning and reduce model size.

Our approach differs from these methods as we do not rely on parameter regularization or expanding architectures. Instead, we focus on preserving the learned representations and their relational structure between tasks through the DWDP loss, which provides a more scalable solution without incurring additional memory overhead.

## A.3 Continual and Multitask Learning

Our work distinguishes itself from existing approaches in CMTL by introducing a new problem domain where new tasks and their associated datasets arrive incrementally. In this setting, the model is not only required to adapt to new tasks but also to utilize these new datasets to enhance its performance on previous tasks. Specifically, when new data for additional tasks becomes available, it is used to further train the existing model. This training process enables the model to reinforce and improve its understanding of prior tasks, effectively allowing it to remember and perform better on both past and current tasks.

Building upon the extensive research in multitask learning (MTL) Caruana (1997); Sener & Koltun (2018); Yu et al. (2020); Navon et al. (2022); Li & Hoiem (2017a) and continual learning (CL) Ratcliff (1990); Robins (1995); Riemer et al. (2018); Aljundi et al. (2019b); Chaudhry et al. (2021); Li & Hoiem (2017a), the emerging field of continual multitask learning seeks to bridge the two paradigms to effectively manage performance across sequential and concurrent tasks Wu et al. (2023).

One of the most related works to ours, Mirzadeh et al. Mirzadeh et al. (2020), focus on the linear mode connectivity between solutions obtained through sequential and simultaneous training. While they demonstrate that a linear path of low error exists for more than twenty tasks and introduce algorithms like Mode Connectivity SGD (MC-SGD) to enhance continual learning, their work does not address the use of *new tasks* to improve performance on previous ones, particularly using a similar setup to traditional continual learning, which means their works fit more on the realm of CL.

Similarly, Liao et al. Liao et al. (2022) propose MUSCLE, a multitask self-supervised continual learning framework designed to pre-train deep models on diverse X-ray datasets. This work, similar to ours, operates in the domain of medical imaging to process classification and segmentation in different body areas. However, their work differs from ours because their focus is on pre-training the model on different tasks for better generalization, which they refer to as "multitask continual learning." We specifically differentiate our CMTL approach from theirs in that our tasks are seen *iteratively*; we do not have access to all tasks at the same time, and the tasks themselves could be orthogonal to previously seen tasks.

In summary, our approach introduces a novel aspect to CMTL by leveraging new tasks and their data not only to learn the new tasks but also to generalize on prior tasks, all within an iterative framework where tasks arrive sequentially and are potentially unrelated. This sets our work apart from existing CMTL methods, which typically do not utilize new tasks to enhance previous ones in this manner.

## B  LEARNING WITH PRESERVING ALGORITHM OVERVIEW

In this section, we present the pseudocode for our algorithm presented in Section 3.

---

**Algorithm 1** Learning with Preserving (LwP)

---

1: **Input:** Sequence of tasks $\{D_t\}_{t=1}^T$, hyperparameters $\lambda_c$, $\lambda_o$, $\lambda_d$
2: **Output:** Final model parameters $\theta^{[T]}$
3: Initialize initial model parameters $\theta^{[0]}$
4: **for** $t = 1$ to $T$ **do**
5:     Initialize current model parameters: $\theta^{[t]} \leftarrow \theta^{[t-1]}$
6:     Add new task-specific layer $g_{\theta_t}$ for task $t$ to $\theta^{[t]}$
7:     Freeze previous model parameters $\theta^{[t-1]}$
8:     **for** each minibatch $\{(\boldsymbol{x}_i, y_i^t)\}_{i=1}^N$ from $D_t$ **do**
9:         Compute shared representations: $\boldsymbol{z}_i^{[t]} = f_{\theta_s^{[t]}}(\boldsymbol{x}_i)$
10:         Compute output for current task: $\hat{y}_i^t = g_{\theta_t^{[t]}}(\boldsymbol{z}_i^{[t]})$
11:         Compute representations from frozen model: $\boldsymbol{z}_i^{[t-1]} = f_{\theta_s^{[t-1]}}(\boldsymbol{x}_i)$
12:         **for** $o = 1$ to $t - 1$ **do**
13:             Compute outputs for previous task $o$:
14:             Current model output: $\hat{y}_i^o = g_{\theta_o^{[t]}}(\boldsymbol{z}_i^{[t]})$
15:             Pseudolabel from frozen model: $\tilde{y}_i^o = g_{\theta_o^{[t-1]}}(\boldsymbol{z}_i^{[t-1]})$
16:         **end for**
17:         Compute loss for new task: $\mathcal{L}_{\text{cur}} \leftarrow \mathcal{L}_{\text{cur}}(y_i^t, \hat{y}_i^t)$
18:         Compute loss for old tasks: $\mathcal{L}_{\text{old}} \leftarrow \sum_{o=1}^{t-1} \mathcal{L}_{\text{old}}(\tilde{y}_i^o, \hat{y}_i^o)$
19:         Compute dynamic mask $m_{ij}$:

$$m_{ij} = \begin{cases} 1, & \text{if } y_i^t = y_j^t, \\ 0, & \text{otherwise} \end{cases}$$

20:         Compute DWDP loss:

$$\mathcal{L}_{\text{DWDP}} \leftarrow \frac{1}{N^2} \sum_{i=1}^N \sum_{j=1}^N m_{ij} \left( d(\boldsymbol{z}_i^{[t-1]}, \boldsymbol{z}_j^{[t-1]}) - d(\boldsymbol{z}_i^{[t]}, \boldsymbol{z}_j^{[t]}) \right)^2$$

21:         Compute total loss:

$$\mathcal{L}_{\text{lwp}} \leftarrow \lambda_c \mathcal{L}_{\text{cur}} + \lambda_o \mathcal{L}_{\text{old}} + \lambda_d \mathcal{L}_{\text{DWDP}}$$

22:         Update parameters $\theta^{[t]}$ by minimizing $\mathcal{L}_{\text{lwp}}$
23:     **end for**
24: **end for**

---

## C  JUSTIFICATION ON USING EUCLIDEAN DISTANCE 3.2

Here, we show that preserving the squared Euclidean distances between the data points in $Z$ and $Z'$ is sufficient to achieve the same effect.

**Squared Euclidean Distance Preservation**   We define the squared Euclidean distance between two points $z_i$ and $z_j$ as:

$$D_{ij}(Z) = \|z_i - z_j\|^2. \tag{17}$$

Similarly, we compute $D_{ij}(Z')$ for $Z'$.

Our goal is to minimize the difference between the squared distances in $Z$ and $Z'$, which we formalize with the following loss function:

$$\mathcal{L}_{\text{dist}}(Z, Z') = \sum_{i=1}^{n} \sum_{j=1}^{n} \left( \|z_i - z_j\|^2 - \|z_i' - z_j'\|^2 \right)^2. \tag{18}$$

Minimizing $\mathcal{L}_{\text{dist}}$ with respect to $Z'$ encourages the squared distances between all pairs of points in $Z'$ to match those in $Z$:

$$\|z_i' - z_j'\|^2 \approx \|z_i - z_j\|^2, \quad \forall i, j. \tag{19}$$

Since the exponential function is Lipschitz continuous on compact subsets, small changes in the squared distance result in small changes in the kernel value. Specifically, if the squared distances are preserved within a small error $\epsilon > 0$:

$$\left| \|z_i - z_j\|^2 - \|z_i' - z_j'\|^2 \right| < \epsilon, \tag{20}$$

then the difference in the kernel values can be bounded:

$$\left| k(z_i, z_j) - k(z_i', z_j') \right| = \left| \exp\left( -\frac{\|z_i - z_j\|^2}{2\sigma^2} \right) - \exp\left( -\frac{\|z_i' - z_j'\|^2}{2\sigma^2} \right) \right| \tag{21}$$

$$\leq \frac{1}{2\sigma^2} \exp\left( -\frac{\min(\|z_i - z_j\|^2, \|z_i' - z_j'\|^2)}{2\sigma^2} \right) \left| \|z_i - z_j\|^2 - \|z_i' - z_j'\|^2 \right| \tag{22}$$

$$\leq \frac{1}{2\sigma^2} \epsilon \tag{23}$$

$$\leq L_k \epsilon \tag{24}$$

where $L_k$ is a Lipschitz constant dependent on $\sigma$.

Therefore, preserving the squared Euclidean distances between $Z$ and $Z'$ implies that the Gaussian kernel matrices $K(Z)$ and $K(Z')$ are approximately equal:

$$k(z_i, z_j) \approx k(z_i', z_j'), \quad \forall i, j. \tag{25}$$

## D  ADDITIONAL DETAILS ON EXPERIMENTAL RESULTS

### D.1  HYPERPARAMETERS

In the following section, we provide an extensive description of the hyperparameters utilized during the training phase. Across all datasets and models, the Adam optimizer Kingma & Ba (2017) was employed universally. For the CelebA and FairFace datasets, a consistent learning rate of 0.0001 was maintained, coupled with a batch size configuration of 256. In contrast, for the PhysiQ dataset, a higher learning rate of 0.01 was utilized alongside a smaller batch size of 32. Furthermore, we adhered to fixed model-specific hyperparameters for all datasets and models to ensure uniformity and consistency, including the LwP parameters. In the case of LwP, the parameters set as follows: $\lambda_n$ as a value of 1, $\lambda_o$ as a value of 1, and $\lambda_d$ with a value of 0.01. Additionally, the 10 tasks used for CelebA are wearing lipsticks, smiling, mouth slightly open, high cheekbones, attractive, heavy makeup, male, young, wavy hair, and straight hair. PhysiQ dataset includes three attributes assessing exercise quality: stability, range of motion, and exercise variation.

Details of all models and their hyperparameter selection have been documented in the codebase. For in-depth understanding and additional information, please consult our code repository available at [ANONYMOUS LINK].

## D.2 COMPARISON WITH MTL METHODS

Table 3: Comparison of Accuracy across Different Models and Datasets

| Method Type | Model | CelebA | PhysiQ | FairFace |
|---|---|---|---|---|
| STL | - | $72.230 \pm 7.297$ | $87.167 \pm 10.102$ | $64.435 \pm 3.660$ |
| MTL | MTL | $\mathbf{76.526 \pm 7.616}$ | $\mathbf{93.536 \pm 5.739}$ | $71.418 \pm 4.169$ |
| | PCGrad | $75.506 \pm 8.146$ | $91.910 \pm 8.491$ | $70.061 \pm 4.892$ |
| | IMTL | $76.280 \pm 7.248$ | $92.661 \pm 6.617$ | $71.399 \pm 3.887$ |
| | NashMTL | $75.506 \pm 8.146$ | $91.518 \pm 7.118$ | $\mathbf{71.607 \pm 3.577}$ |
| CMTL | **LwP** | $73.484 \pm 8.019$ | $88.242 \pm 12.010$ | $68.545 \pm 4.454$ |

All MTL approaches utilize the same model architecture as LwP. Despite being supplied with all labels for every input data point, the amount of training samples for MTL models matches that seen by CL models per task iteration. Aligning with earlier studies, MTL approaches frequently represent the upper bound for all CL models. An interesting discovery is that all MTL models deliver nearly identical performance on the benchmark.

## D.3 ACCURACY PROGRESSION FOR EACH TASK ITERATION

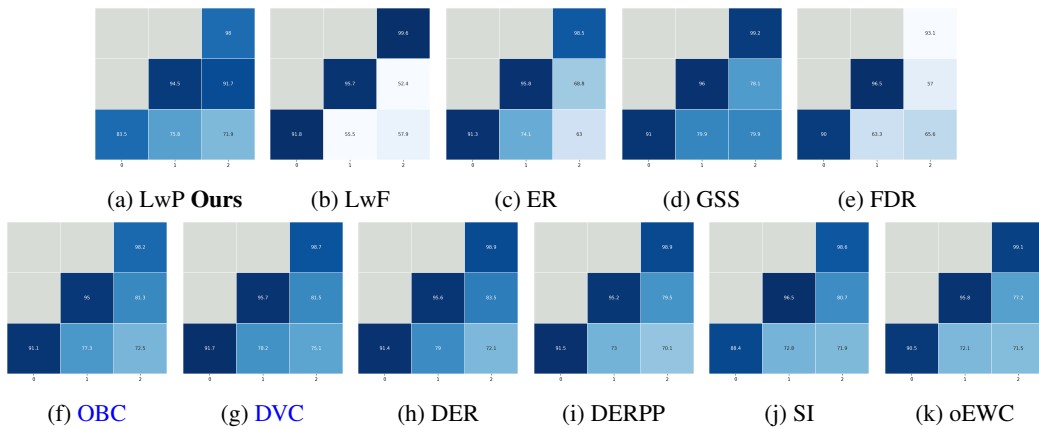

(a) LwP **Ours**    (b) LwF    (c) ER    (d) GSS    (e) FDR

(f) OBC    (g) DVC    (h) DER    (i) DERPP    (j) SI    (k) oEWC

Figure 8: Confusion matrices for different models on the PhysiQ dataset

Figures 8 and 9 illustrate that the application of LwP reduces the issue of catastrophic forgetting in the PhysiQ and FairFace datasets as well. This effect is particularly pronounced when applied to datasets with a large number of samples, such as Fairface and CelebA, in comparison to smaller datasets such as PhysiQ. These observations imply that LwP is a scalable and effective solution to mitigate catastrophic forgetting in continual multitask learning models. We further investigate the effect of the number of training samples on performance in D.4.

## D.4 INFLUENCE OF TRAINING SAMPLE

We include experiment results on the influence of number of training samples to the performance, as shown in Fig. 10a. It shows that our approach outperforms others from 1000 labels and onward, when trained and tested on the PhysiQ dataset.

Fig. 10b illustrates the Expected Calibration Error (ECE) Nixon et al. (2019) for each model in relation to the number of training samples. The ECE quantifies how much confidence a model deviates

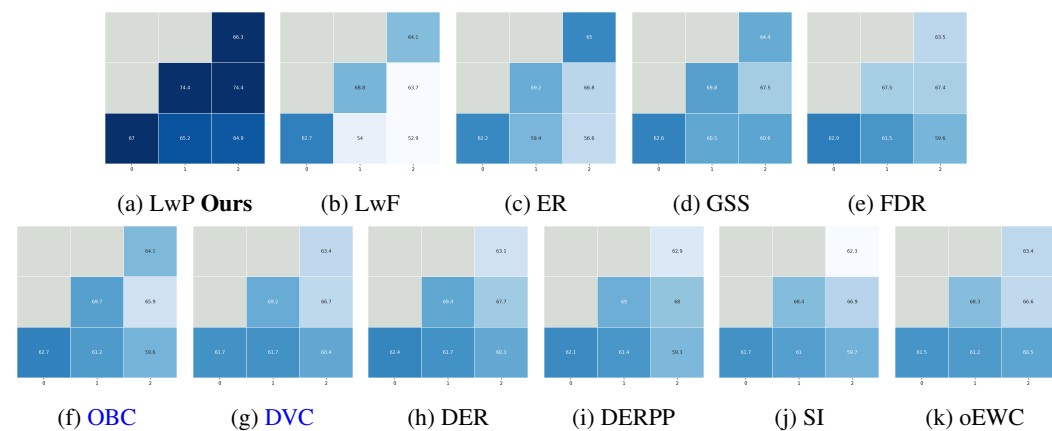

(a) LwP **Ours**     (b) LwF     (c) ER     (d) GSS     (e) FDR

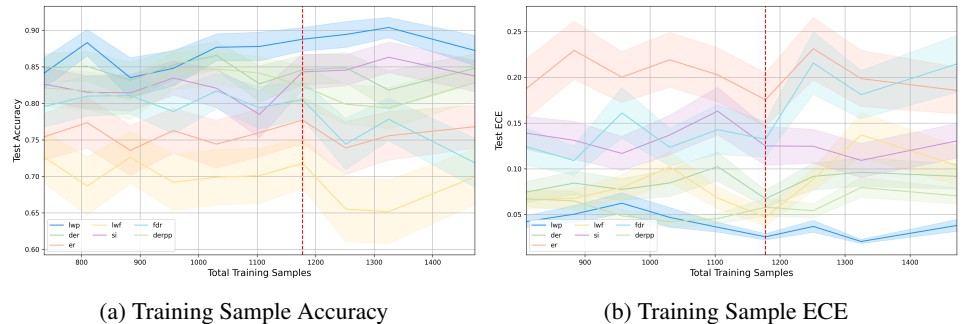

(f) OBC     (g) DVC     (h) DER     (i) DERPP     (j) SI     (k) oEWC

Figure 9: Confusion matrices for different models on the FairFace dataset

(a) Training Sample Accuracy

(b) Training Sample ECE

Figure 10: Training Sample Accuracy and ECE. Noted, the red line represents the buffer size is greater or equal than the batch size, since the buffer size of replay buffer methods is determined by a percentage of the batch size.

from the actual output distribution, with a lower ECE indicating more accurate confidence assignments for a given classification target. This metric is particularly crucial in safety-critical settings, where the model must provide reliable confidence output. As neural network models frequently demonstrate overconfidence Wei et al. (2022), monitoring ECE becomes essential. The figure reveals that LwP not only maintains the lowest variance across different seeds but also achieves the lowest ECE value when the training sample size exceeds roughly 1000.

## D.5 PERFORMANCE IMPROVEMENT PER ITERATION

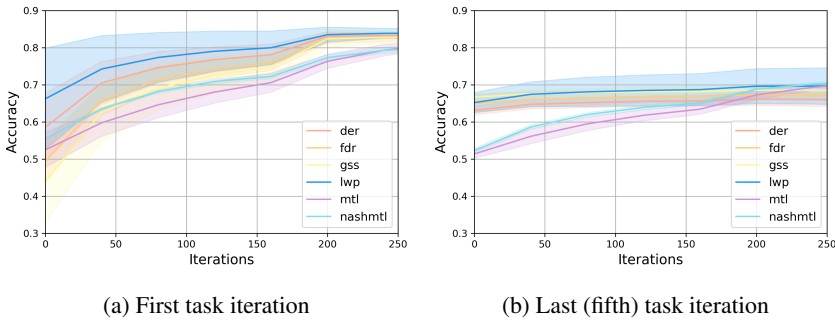

(a) First task iteration

(b) Last (fifth) task iteration

Figure 11: Average accuracy progression per iteration for CelebA with 5 tasks

We show that LwP demonstrates a faster improvement per iteration given the same batch size as other CL and MTL models. Here, we plot the evolution of average accuracy across all tasks seen on the test set over training iteration for the top performing CL and MTL baselines along with LwP. We used the CelebA dataset with 5 task splits. To make a fair comparison with CL models, MTL models were trained on the amount of train data that CL models saw in each iteration with access to all 5 tasks. The accuracies of MTL models are calculated up to what CL models have learned so far.

Fig. 11a shows how quickly each CL and MTL model learns the first task. This can be understood as the speed at which the models acquire knowledge when they have no prior information to "recall". As shown, LwP learns consistently faster per iteration compared to CL and MTL baselines. As MTL models simultaneously learn multiple labels, their convergence per iteration is generally slower compared to CL models in this setting.

Conversely, Fig. 11b illustrates a case where MTL models are trained from the beginning with labels available for all $t$ tasks, whereas CL models, having been pretrained on $t-1$ tasks, must now incrementally learn the $t^{th}$ task while maintaining performance on old tasks. This configuration is crucial in real-world scenarios where the cost of labeling data typically exceeds that of data collection, prompting the decision to gather more partially labeled data rather than re-labeling existing data. Analogous to the prior scenario, the progression of test accuracy over iterations demonstrates that LwP consistently exceeds other CL models and exhibits performance that is competitive with MTL models, which are considered the upper bound for continual learning. This highlights the comparative benefit of LwP when users face the choice between relabeling existing data and obtaining new data with different labels.

### D.6 Effect of Model Parameters and Image Sizes on Training Performance

Table 4: Accuracy Percentage Comparison Across Models on CelebA Dataset

| Method Type | Model | ResNet50 ($32 \times 32$) | ResNet101 ($32 \times 32$) | ResNet50 ($224 \times 224$) |
|---|---|---|---|---|
| CL | LwF | $59.277 \pm 11.920$ | $58.279 \pm 11.202$ | $60.012 \pm 14.448$ |
| | oEWC | $66.975 \pm 10.110$ | $67.159 \pm 10.506$ | $68.511 \pm 13.352$ |
| | ER | $65.335 \pm 9.298$ | $65.646 \pm 8.784$ | $65.973 \pm 14.729$ |
| | SI | $66.698 \pm 10.030$ | $67.456 \pm 9.880$ | $67.747 \pm 13.754$ |
| | GSS | $65.926 \pm 13.120$ | $65.587 \pm 13.142$ | $69.817 \pm 18.771$ |
| | FDR | $61.753 \pm 11.943$ | $61.720 \pm 12.017$ | $65.225 \pm 15.545$ |
| | DER | $62.105 \pm 12.114$ | $63.797 \pm 10.774$ | $69.859 \pm 12.690$ |
| | DERPP | $62.814 \pm 11.071$ | $62.957 \pm 11.577$ | $68.102 \pm 13.557$ |
| | DVC | $67.084 \pm 10.380$ | $65.340 \pm 11.427$ | $70.921 \pm 13.823$ |
| | OBC | $64.220 \pm 11.237$ | $66.058 \pm 10.370$ | $69.319 \pm 13.607$ |
| CMTL | **LwP** | $\mathbf{67.388 \pm 11.125}$ | $\mathbf{69.432 \pm 10.416}$ | $\mathbf{85.064 \pm 5.388}$ |

Table 4 illustrates that the LwP method scales effectively with increased input resolution and model size. We find that preserving the Gaussian kernel, as shown in eq. 3, results in improved performance on larger scales, especially with respect to input resolution. In the ResNet50 benchmark utilizing a 224x224 image size, LwP notably surpasses other baselines by achieving an 85% accuracy, which is about 15% percentage points greater than the runner-up. This suggests that, as the input allows the model to create more insightful representations, LwP becomes increasingly advantageous because it can maintain these representations. We also note that the bigger models with the same input size are not performing as well as the one with resnet18. This is due to the fact that the inputs do not have enough information to capture generalized patterns, resulting in overfitting.

### D.7 Training from MTL to CL

We initially train the model on the first five tasks using a MTL setting, employing ResNet18 as the encoder with input images of size $64 \times 64 \times 3$. After completing the MTL phase, we extract the encoder and freeze its weights. This frozen encoder is then used to train classifiers for the first five tasks in a continual learning CL setting with various models. Subsequently, we train the entire models for the last five tasks under the same CL framework, utilizing the same frozen encoder on the

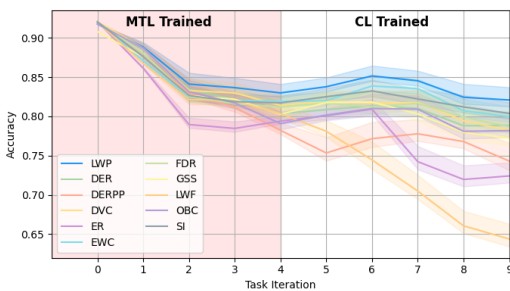

Figure 12: Train the first 5 tasks in MTL setting (meaning all tasks are trained simutaneously using the basic MTL model then applied the encoder into CL models), then we used the encoder to further to train in CL setting with different models with ResNet 18 with an image size of 64.

remaining tasks. This approach allows us to assess the effectiveness of our method in transitioning from MTL to CL while maintaining performance across all tasks.

Our method significantly outperforms other models in terms of accuracy on the CelebA dataset. Specifically, as shown in Table 5, LwP achieves an average accuracy of 83.652%, surpassing all other CL methods tested. The closest competitors, oEWC and SI, attain accuracies of 82.250% and 82.194%, respectively. This demonstrates the effectiveness of our approach in leveraging a MTL pre-trained encoder for subsequent CL tasks.

The superior performance of LwP suggests that initializing the encoder with MTL on the first five tasks provides a robust foundation for learning new tasks in a continual fashion. Our method effectively mitigates catastrophic forgetting by preserving essential features learned during the MTL phase while adapting to new tasks, given the continual tasks are shorter now. This balance between stability and plasticity still allows LwP to maintain high accuracy in the continual learning tasks.

Table 5: Accuracy Percentage Comparison Across Models on CelebA Dataset, Trained on MTL on first 5 tasks then CL on last 5 tasks

| Method Type | Model | ResNet18 ($64 \times 64$) |
|---|---|---|
| | LwF | $74.057 \pm 11.364$ |
| | oEWC | $82.250 \pm 6.362$ |
| | ER | $77.245 \pm 8.434$ |
| | SI | $82.194 \pm 6.460$ |
| CL | GSS | $80.563 \pm 8.239$ |
| | FDR | $81.271 \pm 7.738$ |
| | DER | $81.010 \pm 8.674$ |
| | DERPP | $78.177 \pm 9.532$ |
| | DVC | $81.387 \pm 7.821$ |
| | OBC | $80.516 \pm 8.446$ |
| CMTL | **LwP** | $\mathbf{83.652 \pm 7.069}$ |

Moreover, the lower standard deviation in LwP's performance indicates consistent results across different runs, highlighting the reliability of our approach. The results confirm that combining MTL pre-training with our proposed CL strategy enhances the model's ability to generalize and adapt to new tasks without compromising performance on previously learned tasks.

Similarly in Figure 12, we averaged the results across task iterations to evaluate performance over time. Our method, LwP, demonstrates minimal accuracy loss when training on new tasks, highlighting its performance against forgetting. The standard deviation—represented as 20% of the total for visualization purposes—remains low, indicating consistent performance. Although there is a slight increase in standard deviation during later tasks, suggesting a potential drop in accuracy due to forgetting, LwP still preserves knowledge at a superior level compared to other baselines. Even with the first five tasks trained in a multitask setting, our method maintains the best overall accuracy, outperforming other models in preserving learned information.

