# OpenReview forum: "Learning with Preserving for Continual Multitask Learning"
_ICLR.cc/2025/Conference — Submitted to ICLR 2025_

### Official Review · Reviewer_KdBT · 2024-10-31

**Soundness:** 3
**Presentation:** 2
**Contribution:** 2
**Rating:** 6
**Confidence:** 5

**Summary:**

This paper introduces Learning with Preserving (LwP), a novel approach to continual multitask learning (CMTL) that addresses limitations in traditional continual and multitask learning methods by preserving previously learned knowledge across diverse tasks. LwP employs a Dynamically Weighted Distance Preservation (DWDP) loss function, which maintains representation integrity for both prior and future tasks without relying on a replay buffer.

**Strengths:**

1. The idea is good - multi-task continual learning and is an essential problem in the space of continual learning.
2. The dynamic weighting is an interesting method, but a little ensure if pair-wise comparison is optimal when the dataset is big.
3. Adequate sets of experiments across various metrics.

**Weaknesses:**

1. Why does the authors consider three separate datasets and not a combination of them? The latter would be more representative of real-world scenarios.
Eg: first 3 tasks CelebA, next 3 tasks PhysiQ and so on, which is more representative of a realistic scenario.
2. How is Fig 2 visualized? what exactly it is meant to represent? Is this is a conceptual diagram, a visualization of actual data?

**Questions:**

Please refer to the weakness.

---

> ### Author Response · Authors · 2024-11-16
>
> **Response to Weakness 1:**
>
> We appreciate your suggestion to consider combining datasets, as it could represent more complex and realistic scenarios. However, our focus on individual datasets aligns with the assumptions of the CMTL setting, where the input data distribution remains consistent across tasks and the labels are independent.
>
> Combining datasets like CelebA (image data) and PhysiQ (IMU sensory data) would introduce significant shifts not only in the input distribution but also in the data modality, which falls outside the scope of our current study. Our method is specifically designed to perform optimally under the condition of a consistent input distribution within a single modality, with allowances for minor deviations.
>
> Addressing scenarios that combine multiple modalities, or heterogeneous data would require extending the current framework to handle such complexity. This represents an interesting direction for future work, where our approach could be adapted to more general continual learning scenarios involving multimodal datasets.
>
> **Response to Weakness 2:**
>
> Thank you for bringing this to our attention. Figure 2 is a conceptual diagram designed to illustrate the framework of our proposed LwP method. The images depicted are examples from the datasets (e.g., cancer tissues) and are included to convey the idea that the model is continuously learning different attributes (tasks) over time.
>
> We acknowledge that the caption and explanation of Figure 2 could be clearer.  In the revised manuscript, we will provide a more detailed description to clarify its purpose and ensure it effectively communicates the intended concept to the readers.

---

> ### Author Response · Authors · 2024-11-25
> **To Verify if Our Responses Have Addressed Your Concerns and Express Our Gratitude**
>
> Dear Reviewer,
>
> We deeply value the time and effort you have dedicated to reviewing our paper and providing insightful suggestions. As the discussion phase is coming to an end and no further author-reviewer interactions are planned, we would like to confirm if our responses from this and a few days ago have successfully addressed your concerns. We hope we have resolved the issues raised. However, if there are any points that require further clarification or additional concerns you would like us to address, please feel free to reach out. We remain fully committed to continuing our discussion with you.
>
> Best regards.

---

> > ### Comment · Reviewer_KdBT · 2024-12-02
> >
> > Thank you for your clarification. I would like to hold to my rating of 6 (Marginal accept)

---

### Official Review · Reviewer_9JHi · 2024-11-02

**Soundness:** 1
**Presentation:** 2
**Contribution:** 2
**Rating:** 5
**Confidence:** 3

**Summary:**

The paper aims to address the continual multi task problem. The paper proposes a LwP loss in addiction to current loss and loss to preserve old preditions. LwP tries to preserve the knowledge in the implicit knowlege space. The paper also propose to masks the loss on LwP if the labels are different and in that case it is not nessory to have this preserving loss.

The paper then goes to evaulate this approch on various small scale benchmarks, and specilay on image datasets, it shows a clear gains over previous approches. The paper also shows the BWT metric for all the continual learning methods, and t-sne plots for the latent space.

**Strengths:**

Performance on all the benchmarks are impressive. Figure 5 clearly shows the minimal loss in performance in the previous tasks as the learning progress.

the benifits of Learning with Preserving (LwP) loss as a regulaization is very solid, and can be seen on the figure 5 and table 1, and compared to other appoches LwP performs considerably well.

the evaulation is done with a good coverage, with 3 vision benchmarks, and show the distributions of these latents in t-sne plots. the paper also measures the backward transfer values of the continual learning methods.

**Weaknesses:**

It is not clear, how this CMTL problem is novel, it is same as in early LwF papers, and the paper claims this is one of the contibutions. please adress this in the rebuttal.

while the results are impressive, i am bit scaptical on the scale of the datasets, all have been trained on smaller scale and low resolution. would be nice to show some results on larger resolution images and models. Also would be nice to show that this approch can work for other archituctres like vit. I belive it should work without any problems. I still think resnet 18 is too small model in the current landscape to validate anything concretely.

Also there is not enough ablations to varify the contibutions of dynamic weighting, that would be helpful to validate this claim.

**Questions:**

please look at my strengths and weakness sections, and if you can adress the weakness section, i am happy to change my ratings.

---

> ### Author Response · Authors · 2024-11-16
>
> **Response to Weakness 1:**
>
> In the CMTL setting, we focus on scenarios where the labels represent independent attributes of the same input domain shared across time. For example, using the CelebA dataset, one-third of the data may focus on learning one attribute (e.g., "smiling"), another third on a different attribute (e.g., "wearing glasses"), and the final third on yet another attribute (e.g., "gender"). Importantly, these new label tasks can also apply to data from previous tasks, making this setting distinct.
>
> This introduces unique challenges that differ from earlier works like LwF. While LwF typically addresses scenarios involving mutually exclusive labels or data distribution shifts, CMTL operates under the stricter condition of consistent input distributions across tasks. This requires models to effectively utilize shared input distributions while handling independent labels, a setting where existing methods often struggle to outperform single-task learning baselines.
>
> Our contribution lies in addressing this gap with a specialized approach tailored to the CMTL setting, enabling effective knowledge preservation and multitask learning under these more realistic and challenging conditions.
>
>
> **Response to Weakness 2:**
>
> Thank you for raising this important point. Our experiments currently cover datasets with varying scales—CelebA with 10 tasks at 32×32 resolution, FairFace at standard resolutions using ResNet-18, and PhysiQ as a time-series benchmark. We agree that testing our method on higher-resolution images and larger models would provide additional evidence of its effectiveness.
>
> Since our approach is not constrained by the architecture, as long as there is a representation space before the output layer, we are confident it can generalize to other models like ViT . To strengthen our validation and demonstrate the scalability and robustness of our method in more demanding settings, we will conduct further experiments with larger models and higher-resolution datasets and include these results in the revised paper.
>
> **Response to Weakness 3:**
>
> Thank you for pointing out the importance of further validating the contribution of dynamic weighting. We have addressed this aspect in our ablation study (Appendix E.5). To briefly summarize, we evaluated the impact of our proposed loss function by selectively disabling the dynamic weighting feature and comparing it with other structure-preserving loss functions. The baselines included in our assessment are CO2L [Cha et al., 2021], RKD [Park et al., 2019], cosine similarity, and the RBF kernel [Han et al., 2012].
>
> The results, presented in Table 3, show that the loss function with both dynamic weighting and Euclidean distance consistently outperforms these alternatives. This highlights the importance of each component in achieving optimal performance. We believe that the superior performance of Euclidean distance with dynamic weighting is due to its unnormalized nature across batches, unlike previously proposed methods.
>
> We will ensure this point is emphasized more clearly in the revised manuscript to underline the contribution of dynamic weighting.
>
> **Reference:**
>
> Hyuntak Cha, Jaeho Lee, and Jinwoo Shin. Co2L: Contrastive Continual Learning, June 2021. URL http://arxiv.org/abs/2106.14413. arXiv:2106.14413 [cs].
>
> Wonpyo Park, Dongju Kim, Yan Lu, and Minsu Cho. Relational Knowledge Distillation, May 2019. URL http://arxiv.org/abs/1904.05068. arXiv:1904.05068 [cs].
>
> Shunjie Han, Cao Qubo, and Han Meng. Parameter selection in svm with rbf kernel function. In World Automation Congress 2012, pp. 1–4. IEEE, 2012.

---

> ### Author Response · Authors · 2024-11-25
> **Regarding Concerns of Model Size and Image Resolution and Ablation Study**
>
> ### Response to the concern of the performance of model size and image resolution.
>
> We have expanded our experiments to assess the applicability of LwP in bigger models and higher-resolution of the dataset, CelebA. We have conducted experiments using ResNet-50 and ResNet-101 models on 64x64 and 224x224 resolution datasets. But ViT was omitted as the architecture demands large samples for effective training and is relatively much more sensitive to optimizer hyperparameters compared to ResNets. The results are included in the appendix to provide a more comprehensive validation of our approach.
>
> | **Model**    | **ResNet50 (32×32)** | **ResNet101 (32×32)** | **ResNet50 (224×224)** |
> |--------------|----------------------|-----------------------|-------------------------|
> | LwF          | 59.277 ± 11.920     | 58.279 ± 11.202      | 60.012 ± 14.448        |
> | oEWC         | 66.975 ± 10.110     | 67.159 ± 10.506      | 68.511 ± 13.352        |
> | ER           | 65.335 ± 9.298      | 65.646 ± 8.784       | 65.973 ± 14.729        |
> | SI           | 66.698 ± 10.030     | 67.456 ± 9.880       | 67.747 ± 13.754        |
> | GSS          | 65.926 ± 13.120     | 65.587 ± 13.142      | 69.817 ± 18.771        |
> | FDR          | 61.753 ± 11.943     | 61.720 ± 12.017      | 65.225 ± 15.545        |
> | DER          | 62.105 ± 12.114     | 63.797 ± 10.774      | 69.859 ± 12.690        |
> | DERPP        | 62.814 ± 11.071     | 62.957 ± 11.577      | 68.102 ± 13.557        |
> | DVC          | 67.084 ± 10.380     | 65.340 ± 11.427      | 70.921 ± 13.823        |
> | OBC          | 64.220 ± 11.237     | 66.058 ± 10.370      | 69.319 ± 13.607        |
> | **LwP (Ours)**      | **67.388 ± 11.125** | **69.432 ± 10.416**  | **85.064 ± 5.388**     |
>
> This demonstrates the competitiveness of our approach. We have included these results in the revised manuscript to provide a more comprehensive evaluation of LwP's performance in continual multitask learning scenarios in the Appendix D.6. We thank the reviewer again for highlighting this point.
>
> ### Response to the Ablation Study
> To strengthen our work and provide additional clarity, we have moved the ablation study from the appendix to the main paper. For further clarification, the results are also presented here for reference.
>
>
> | **Method on PhysiQ**       | **LwP ($L^2$)**      | **LwP (Cosine)**      | **LwP (RBF)**         | **IRD (Co2L)**       | **RKD**            |
> |----------------------------|----------------------|-----------------------|-----------------------|----------------------|--------------------|
> | **Dynamic Weighting**      | **88.2 ± 12.0**      | 85.4 ± 13.1           | 84.5 ± 13.7           | 86.4 ± 11.5          | 85.1 ± 13.3        |
> | **W/o Dynamic Weighting**  | 86.0 ± 12.3          | 84.1 ± 14.4           | 84.8 ± 14.5           | 79.9 ± 17.1          | 85.9 ± 11.9        |
>
> The ablation study demonstrates that LwP using $L^2$ with dynamic weighting outperforms other variations and baselines. We have included these results in the revised manuscript to provide a more comprehensive evaluation of LwP's performance in continual multitask learning scenarios in Section 4.6. We thank the reviewer for highlighting this point.

---

> ### Author Response · Authors · 2024-11-25
> **To Verify if Our Responses Have Addressed Your Concerns and Express Our Gratitude**
>
> Dear Reviewer,
>
> We deeply value the time and effort you have dedicated to reviewing our paper and providing insightful suggestions. As the discussion phase is coming to an end and no further author-reviewer interactions are planned, we would like to confirm if our responses from this and a few days ago have successfully addressed your concerns. We hope we have resolved the issues raised. However, if there are any points that require further clarification or additional concerns you would like us to address, please feel free to reach out. We remain fully committed to continuing our discussion with you.
>
> Best regards.

---

### Official Review · Reviewer_minZ · 2024-11-03

**Soundness:** 2
**Presentation:** 2
**Contribution:** 2
**Rating:** 6
**Confidence:** 4

**Summary:**

This paper introduces Learning with Preserving (LwP), a novel framework designed for Continual Multitask Learning (CMTL), which involves learning different tasks sequentially while preserving shared representations. The paper evaluates LwP on three benchmark datasets across two modalities, demonstrating its competitive performance compared to existing continual learning methods.

**Strengths:**

1.	The paper proposes a new scenario of continual learning, CMTL, highlighting its unique challenges and significance in practical applications.
2.	The LwP framework is innovative in preserving previously learned knowledge in a way that remains applicable and beneficial across diverse tasks.
3.	The experimental results suggest that LwP demonstrates competitive performance compared to existing continual learning methods.

**Weaknesses:**

1.	How does the proposed method address the fundamental challenges in continual learning, such as catastrophic forgetting or the stability-plasticity dilemma?
2.	The Dynamically Weighted Distance Preservation (DWDP) loss is an innovative contribution. However, it would be valuable to delve deeper into the theoretical foundations of DWDP, exploring its relationship to other distance-preserving techniques and providing additional insights into why it is effective for preserving implicit knowledge.
3.	A point of concern is that, continuous learning methods in the comparison experiment are not state-of-the-art, and therefore may not effectively substantiate the validity of the method proposed in this paper.
4.	Further exploration is needed for more experimental settings, such as investigating the performance of a model when continuously learning five tasks in the presence of five base tasks.
5.	The section on the extension to learning problems (pages 19-20) provides a valuable insight into the theoretical underpinnings of LwP, but it could be integrated more seamlessly into the main body of the paper to enhance its readability and coherence.

**Questions:**

See weakness

---

> ### Author Response · Authors · 2024-11-16
>
> **Response to Weakness 1:**
>
> Our proposed method addresses catastrophic forgetting in the CMTL setting through the Dynamically Weighted Distance Preservation (DWDP) loss. This loss function is aimed to preserves approximate solutions for any problems that can be defined in z by leveraging the universality of kernel machines with the Gaussian kernel as approximators (see Sec. 3.2).     By maintaining the integrity of the shared representation space, the DWDP loss enables the model to retain valuable knowledge from prior tasks while effectively learning new ones.
>
> Moreover, our approach inherently balances the stability-plasticity dilemma. By employing dynamic weights, it prioritizes the acquisition of new information without destabilizing previously acquired knowledge. This ensures a harmonious trade-off between stability (preserving past knowledge) and plasticity (adapting to new knowledge), which is critical for success in continual learning scenarios.
>
> **Response to Weakness 2:**
>
> We agree that a deeper theoretical exploration of the DWDP loss would enhance the paper. While our work provides initial analysis and ablation studies (see Appendix E.5) comparing DWDP with other distance-preserving techniques, we found that using the unnormalized Euclidean distance effectively preserves the global structure of the representation space, resulting in improved performance. To briefly summarize, we evaluated the impact of our proposed loss function by selectively disabling the dynamic weighting feature and comparing it with other structure-preserving loss functions. The baselines included in our assessment are CO2L [Cha et al., 2021], RKD [Park et al., 2019], cosine similarity, and the RBF kernel [Han et al., 2012].
>
> We hypothesize that this is due to the unnormalized distances capturing absolute relationships between representations more accurately. To strengthen the paper, we will expand the theoretical discussion in the revised version.
>
> **Response to Weakness 3:**
>
> Thank you for your feedback regarding the methods used for comparison. We understand the importance of evaluating our approach against current and relevant baselines, and we did our best to utilize well-established methods. In our study, we included Dark Experience Replay (DER) and its enhanced version, DER++, as part of our comparative analysis [Buzzega et al., 2020], as they are publicly available and well-documented in the literature. These methods are not only widely used in continual learning scenarios but have also been utilized as strong baselines in other settings, as demonstrated in recent works [Fostiropoulos et al., 2023; Kim et al., 2023]. This reinforces their relevance to our comparative analysis.
>
> DER and DER++ are robust baselines in continual learning scenarios, offering strong performance and relevance to our research setting. By incorporating these methods, we ensured a comprehensive and up-to-date evaluation of our approach.
>
> If you have specific methods in mind that could further strengthen our comparative analysis, we are open to including them to provide a more thorough and well-rounded evaluation.

---

> ### Author Response · Authors · 2024-11-16
>
> Continue to our previous comment
>
> **Response to Weakness 4:**
>
> We appreciate your suggestion to explore the performance of the model when continuously learning additional tasks beyond the initial base tasks. This is an important scenario to evaluate the scalability and robustness of our approach. We will conduct further experiments in this direction and include the results in the revised paper to provide a more comprehensive validation of our method.
>
> **Response to Weakness 5:**
>
> Thank you for your valuable feedback. We agree that integrating the theoretical insights on the extension to learning problems more seamlessly into the main body would improve the paper's readability and coherence. In the revised manuscript, we will restructure the content to incorporate this section into the main text, ensuring a smoother narrative flow and better alignment with the rest of the paper.
>
> **References:**
>
> Buzzega, Pietro, et al. "Dark experience for general continual learning: a strong, simple baseline." Advances in Neural Information Processing Systems 33 (2020): 15920-15930.
>
> Fostiropoulos, Iordanis, Jiaye Zhu, and Laurent Itti. "Batch model consolidation: A multi-task model consolidation framework." Proceedings of the IEEE/CVF Conference on Computer Vision and Pattern Recognition. 2023.
>
> Kim, Sanghwan, et al. "Achieving a better stability-plasticity trade-off via auxiliary networks in continual learning." Proceedings of the IEEE/CVF Conference on Computer Vision and Pattern Recognition. 2023.
>
> Hyuntak Cha, Jaeho Lee, and Jinwoo Shin. Co2L: Contrastive Continual Learning, June 2021. URL http://arxiv.org/abs/2106.14413. arXiv:2106.14413 [cs].
>
> Wonpyo Park, Dongju Kim, Yan Lu, and Minsu Cho. Relational Knowledge Distillation, May 2019. URL http://arxiv.org/abs/1904.05068. arXiv:1904.05068 [cs].
>
> Shunjie Han, Cao Qubo, and Han Meng. Parameter selection in svm with rbf kernel function. In World Automation Congress 2012, pp. 1–4. IEEE, 2012.

---

> > ### Author Response · Authors · 2024-11-20
> > **Regarding Weakness 3 on baselines**
> >
> > Dear Reviewer minZ,
> >
> > Thank you for your feedback and concern regarding the baselines. In response, we revisited state-of-the-art methods to ensure a comprehensive comparative analysis. Below, we clarify the scope of our work and the rationale for selecting the baselines.
> >
> > Our study focuses on lightweight, non-pretrained models to emphasize theoretical contributions and practical applications across diverse modalities. Pretrained models are typically unavailable or unsuitable for continual multitask learning in diverse modalities like IMU sensing, which is one focus of this work.
> >
> > To address your concerns, we categorized existing works into two groups:
> >
> > 1. **Theoretical advancements for performance enhancement without external data (online method).**
> >    In this category, we incorporated *Dual View Consistency (DVC)* [Gu et al., 2022] and *Online Bias Correction (OBC)* [Chrysakis and Moens, 2023] into our comparative analysis. DVC and OBC represent a recent and relevant approach in online class-incremental continual learning, strengthening the validation of our method.
> >
> >    We have updated the manuscript to include the results of DVC and OBS in the comparative tables and figures. As shown below, our proposed method, *LwP*, consistently outperforms DVC and OBS:
> >
> >    | **Method**      | **CelebA (10 tasks)** | **PhysiQ (3 tasks)**   | **Fairface (3 tasks)**   |
> >    |------------------|-----------------------|------------------------|--------------------------|
> >    | **DVC**         | 71.441 ± 7.640        | 85.100 ± 10.381        | 63.848 ± 3.193           |
> >    | **OBC**         | 70.829 ± 8.267        | 83.999 ± 11.377        | 63.872 ± 3.449          |
> >    | **LwP (Ours)**  | **73.484 ± 8.019**    | **88.242 ± 12.010**    | **66.482 ± 3.138**       |
> >
> >    This addition demonstrates the competitiveness of our approach, even against a strong and recently proposed state-of-the-art baseline. We thank the reviewer for highlighting this point and have revised the manuscript accordingly.
> >
> > 2. **Works on continual learning that fall outside the scope of our approach (offline).**
> >    While several continual learning approaches exist in the literature, many are not directly comparable to our method due to differences in assumptions or objectives. For instance, pretrained models are currently beyond the scope of our work, as our emphasis is on developing approaches that operate without large-scale external data or pretrained foundations. However, we recognize the potential compatibility of pretrained models with our framework and will explore this integration in future extensions.
> >    Though there are several other works on Continual Learning [Zhang et al., 2023] and [ Wang et al., 2024] in the past several years, but they are not compatible to be compared with our approach because they are operating on a large scale of external data or pretrained foundations. While these models are currently outside the scope of our work, we acknowledge their potential compatibility with our framework and may explore how to integrate this in future extensions.
> >
> > We believe the baselines included in our paper are sufficient, as they encompass a diverse set of algorithmic methods, including approaches focused on replay buffers [Buzzega et al., 2020], latent alignment (e.g., GSS) [Aljundi et al., 2019], and feature disentanglement regularization (FDR) [Benjamin et al., 2018], and are well-aligned with the continual multitask learning domain.
> >
> > **Reference**
> >
> > Gu, Yanan, et al. "Not just selection, but exploration: Online class-incremental continual learning via dual view consistency." *Proceedings of the IEEE/CVF Conference on Computer Vision and Pattern Recognition.* 2022.
> >
> > Chrysakis, Aristotelis, and Marie-Francine Moens. "Online bias correction for task-free continual learning." ICLR 2023 at OpenReview (2023).
> >
> > Zhang, Gengwei, et al. "Slca: Slow learner with classifier alignment for continual learning on a pre-trained model." Proceedings of the IEEE/CVF International Conference on Computer Vision. 2023.
> >
> > Wang, Liyuan, et al. "Hierarchical decomposition of prompt-based continual learning: Rethinking obscured sub-optimality." Advances in Neural Information Processing Systems 36 (2024).
> >
> > Aljundi, Rahaf, et al. "Gradient based sample selection for online continual learning." Advances in neural information processing systems 32 (2019).
> >
> > Benjamin, Ari S., David Rolnick, and Konrad Kording. "Measuring and regularizing networks in function space." arXiv preprint arXiv:1805.08289 (2018).
> >
> > Buzzega, Pietro, et al. "Dark experience for general continual learning: a strong, simple baseline." Advances in neural information processing systems 33 (2020): 15920-15930.

---

> ### Author Response · Authors · 2024-11-25
> **Regarding Experiment of MTL to CL**
>
> ### Response to the concern of the performance of a model when continuously learning five tasks in the presence of five base tasks.
>
> We have conducted additional experiments to assess the applicability of LwP in continual multitask learning scenarios where the model continuously learns additional tasks after initially learning five tasks. The model was tested on five base tasks and five additional tasks in a multitask learning manner on the CelebA dataset. The results are presented in the table below:
>
> | **Model**    | **ResNet18 (64×64)**  |
> |--------------|-----------------------|
> | LwF          | 74.057 ± 11.364      |
> | oEWC         | 82.250 ± 6.362       |
> | ER           | 77.245 ± 8.434       |
> | SI           | 82.194 ± 6.460       |
> | GSS          | 80.563 ± 8.239       |
> | FDR          | 81.271 ± 7.738       |
> | DER          | 81.010 ± 8.674       |
> | DERPP        | 78.177 ± 9.532       |
> | DVC          | 81.387 ± 7.821       |
> | OBC          | 80.516 ± 8.446       |
> | **LwP (Ours)**      | **83.652 ± 7.069**   |
>
> The table presents the performance of various methods when continuously learning five tasks in the presence of five base tasks using ResNet18 with a 64×64 resolution. We show that LwP outperforms the other methods, demonstrating its effectiveness in handling continual multitask learning scenarios. These results have been included in the revised manuscript to provide a more comprehensive evaluation of LwP's performance in continual multitask learning scenarios in Appendix D.7, along with a diagram illustrating the experiment in a different perspective. We thank the reviewer for highlighting this point.

---

> ### Author Response · Authors · 2024-11-25
> **To Verify if Our Responses Have Addressed Your Concerns and Express Our Gratitude**
>
> Dear Reviewer,
>
> We deeply value the time and effort you have dedicated to reviewing our paper and providing insightful suggestions. As the discussion phase is coming to an end and no further author-reviewer interactions are planned, we would like to confirm if our responses from this and a few days ago have successfully addressed your concerns. We hope we have resolved the issues raised. However, if there are any points that require further clarification or additional concerns you would like us to address, please feel free to reach out. We remain fully committed to continuing our discussion with you.
>
> Best regards.

---

### Official Review · Reviewer_qi2x · 2024-11-06

**Soundness:** 3
**Presentation:** 3
**Contribution:** 2
**Rating:** 3
**Confidence:** 5

**Summary:**

This paper introduces a new problem setting called Continual Multitask Learning (CMTL) and proposes a novel method called Learning with Preserving (LwP) to address it. CMTL is defined as a scenario where a model needs to learn multiple different tasks sequentially, with input data coming from the same distribution but each task having distinct label spaces. The proposed LwP method aims to preserve previously learned knowledge in the shared representation space without requiring a replay buffer of old data. It uses a novel Dynamically Weighted Distance Preservation (DWDP) loss to maintain the integrity of representations. Extensive experiments demonstrate LwP's strong performance and generalization abilities in CMTL scenarios.

**Strengths:**

1. A new continual learning setting is introduced.
2. A method tailored to the new setting is designed.

**Weaknesses:**

1. Compared to general CL scenarios, such as CIL, DIL (domainincremental learning), TIL (task incremental learning), the proposed CMTL setting can indeed be seen as an idealized simplified version. CMTL lets the input data come from the same distribution, which means that all tasks are performed on the same data domain, without considering the case where the data distribution drifts over time. In real-world applications, the data distribution of subsequent tasks may differ from the previous ones.
2. It is difficult to imagine how this setting could be implemented in reality. In actual scenarios, it might only be achievable by repeatedly labeling the same set of data with new labels. Even updating the data slightly would likely change its domain distribution
3. The methods used for comparison are somehow out of date.
4.The performance of LwP likely depends on careful tuning of the loss weights (λc, λo, λd).

**Questions:**

Please refer to my comments in the 'Weakness' session.

---

> ### Author Response · Authors · 2024-11-16
>
> **Response to Weakness 1:**
>
> Thank you for your thoughtful feedback.
>
> Our goal is to leverage the continual multitask learning (CMTL) framework to develop a generalized representation space that consistently outperforms single-task learning (STL) and other baseline methods.
>
> In one of our evaluation settings, we treat each subset of the dataset, defined by a specific label, as a separate task. While there is no data distribution shift in this setup, the key challenge lies in effectively identifying the current feature embeddings and aligning them with the requirements of new tasks, ensuring seamless integration and preservation of prior knowledge.
>
> **Response to Weakness 2:**
>
> CMTL assumes that data for different tasks are sampled from the same overall distribution but are not necessarily the same data points. This reflects real-world scenarios where different tasks involve related data domains without requiring repeated labeling of the exact same data.
>
> For example, consider datasets collected from U.S. roads for different purposes: one for pedestrian detection and another for lane marking detection, collected sequentially. While the specific images may differ, they share common characteristics due to being from the same environment. Similarly, in the medical domain, the same MRI scan could be labeled sequentially by different professionals. A primary care physician might annotate the scan from one perspective, while a radiologist provides a secondary annotation with a specialized focus. This process mirrors scenarios where tasks involve related but distinct labeling objectives, contributing to a more comprehensive understanding of the data [Freeman et al., 2021].
>
> In our experiments, we ensured that the data for each task was unique to that task, reflecting practical scenarios like the examples above. We will revise the paper to better articulate this point and avoid potential misunderstandings.
>
> **Response to Weakness 3:**
>
> Thank you for your feedback regarding the methods used for comparison. We understand the importance of evaluating our approach against current and relevant baselines, and we did our best to utilize well-established methods. In our study, we included Dark Experience Replay (DER) and its enhanced version, DER++, as part of our comparative analysis [Buzzega et al., 2020], as they are publicly available and well-documented in the literature. These methods are not only widely used in continual learning scenarios but have also been utilized as strong baselines in other settings, as demonstrated in recent works [Fostiropoulos et al., 2023; Kim et al., 2023]. This reinforces their relevance to our comparative analysis.
>
> DER and DER++ are robust baselines in continual learning scenarios, offering strong performance and relevance to our research setting. By incorporating these methods, we ensured a comprehensive and up-to-date evaluation of our approach. If you have specific methods in mind that could further strengthen our comparative analysis, we are open to including them to provide a more thorough and well-rounded evaluation
>
> **Response to Weakness 4:**
>
> We appreciate your concern about hyperparameter tuning. In our experiments, we used the same loss weights (λ_c, λ_o, λ_d) across all three datasets. This consistency demonstrates that our method is robust and does not require extensive hyperparameter adjustments for different tasks or datasets.
>
> Additionally, we followed the parameter usage design outlined in the original papers, ensuring that our methodology is grounded in well-established practices. We will revise the paper to emphasize the stability, practicality, and reliability of our approach, further highlighting this important aspect of our work.
>
> **References:**
>
> Freeman, Beverly, et al. "Iterative quality control strategies for expert medical image labeling." Proceedings of the AAAI Conference on Human Computation and Crowdsourcing. Vol. 9. 2021.
>
> Buzzega, Pietro, et al. "Dark experience for general continual learning: a strong, simple baseline." Advances in Neural Information Processing Systems 33 (2020): 15920-15930.
>
> Fostiropoulos, Iordanis, Jiaye Zhu, and Laurent Itti. "Batch model consolidation: A multi-task model consolidation framework." Proceedings of the IEEE/CVF Conference on Computer Vision and Pattern Recognition. 2023.
>
> Kim, Sanghwan, et al. "Achieving a better stability-plasticity trade-off via auxiliary networks in continual learning." Proceedings of the IEEE/CVF Conference on Computer Vision and Pattern Recognition. 2023.

---

> > ### Author Response · Authors · 2024-11-20
> > **Regarding Weakness 3 on baselines**
> >
> > Dear Reviewer qi2x,
> >
> > Thank you for your comments regarding the baselines. In response, we conducted an additional review of state-of-the-art methods to ensure a comprehensive comparative analysis. Below, we provide further clarification on the scope of our work and the rationale behind our choice of baselines.
> >
> > Our focus lies on lightweight, un-pretrained models, as our work emphasizes theoretical contributions and practical applications across different modalities without reliance on pretrained foundational models or large-scale architectures. Such models are typically unavailable for continual multitask learning in diverse modalities (e.g., IMU sensing, as applied in this paper).
> >
> > To address your feedback, we categorized existing works into two groups:
> >
> > 1. **Theoretical advancements for performance enhancement without external data (online method).**
> >    In this category, we incorporated *Dual View Consistency (DVC)* [Gu et al., 2022] and *Online Bias Correction (OBC)* [Chrysakis and Moens, 2023] into our comparative analysis. DVC and OBC represent a recent and relevant approach in online class-incremental continual learning, strengthening the validation of our method.
> >
> >    We have updated the manuscript to include the results of DVC and OBS in the comparative tables and figures. As shown below, our proposed method, *LwP*, consistently outperforms DVC and OBS:
> >
> >    | **Method**      | **CelebA (10 tasks)** | **PhysiQ (3 tasks)**   | **Fairface (3 tasks)**   |
> >    |------------------|-----------------------|------------------------|--------------------------|
> >    | **DVC**         | 71.441 ± 7.640        | 85.100 ± 10.381        | 63.848 ± 3.193           |
> >    | **OBC**         | 70.829 ± 8.267        | 83.999 ± 11.377        | 63.872 ± 3.449          |
> >    | **LwP (Ours)**  | **73.484 ± 8.019**    | **88.242 ± 12.010**    | **66.482 ± 3.138**       |
> >
> >    This addition demonstrates the competitiveness of our approach, even against a strong and recently proposed state-of-the-art baseline. We thank the reviewer for highlighting this point and have revised the manuscript accordingly.
> >
> > 2. **Works on continual learning that fall outside the scope of our approach (offline).**
> >    While several continual learning approaches exist in the literature, many are not directly comparable to our method due to differences in assumptions or objectives. For instance, pretrained models are currently beyond the scope of our work, as our emphasis is on developing approaches that operate without large-scale external data or pretrained foundations. However, we recognize the potential compatibility of pretrained models with our framework and will explore this integration in future extensions.
> >    Though there are several other works on Continual Learning [Zhang et al., 2023] and [ Wang et al., 2024] in the past several years, but they are not compatible to be compared with our approach because they are operating on a large scale of external data or pretrained foundations. While these models are currently outside the scope of our work, we acknowledge their potential compatibility with our framework and may explore how to integrate this in future extensions.
> >
> > We believe the baselines included in our paper are sufficient, as they encompass a diverse set of algorithmic methods, including approaches focused on replay buffers [Buzzega et al., 2020], latent alignment (e.g., GSS) [Aljundi et al., 2019], and feature disentanglement regularization (FDR) [Benjamin et al., 2018], and are well-aligned with the continual multitask learning domain.
> >
> > **Reference**
> >
> > Gu, Yanan, et al. "Not just selection, but exploration: Online class-incremental continual learning via dual view consistency." *Proceedings of the IEEE/CVF Conference on Computer Vision and Pattern Recognition.* 2022.
> >
> > Chrysakis, Aristotelis, and Marie-Francine Moens. "Online bias correction for task-free continual learning." ICLR 2023 at OpenReview (2023).
> >
> > Zhang, Gengwei, et al. "Slca: Slow learner with classifier alignment for continual learning on a pre-trained model." Proceedings of the IEEE/CVF International Conference on Computer Vision. 2023.
> >
> > Wang, Liyuan, et al. "Hierarchical decomposition of prompt-based continual learning: Rethinking obscured sub-optimality." Advances in Neural Information Processing Systems 36 (2024).
> >
> > Aljundi, Rahaf, et al. "Gradient based sample selection for online continual learning." Advances in neural information processing systems 32 (2019).
> >
> > Benjamin, Ari S., David Rolnick, and Konrad Kording. "Measuring and regularizing networks in function space." arXiv preprint arXiv:1805.08289 (2018).
> >
> > Buzzega, Pietro, et al. "Dark experience for general continual learning: a strong, simple baseline." Advances in neural information processing systems 33 (2020): 15920-15930.

---

> ### Author Response · Authors · 2024-11-25
> **To Verify if Our Responses Have Addressed Your Concerns and Express Our Gratitude**
>
> Dear Reviewer,
>
> We deeply value the time and effort you have dedicated to reviewing our paper and providing insightful suggestions. As the discussion phase is coming to an end and no further author-reviewer interactions are planned, we would like to confirm if our responses from this and a few days ago have successfully addressed your concerns. We hope we have resolved the issues raised. However, if there are any points that require further clarification or additional concerns you would like us to address, please feel free to reach out. We remain fully committed to continuing our discussion with you.
>
> Best regards.

---

### Author Response · Authors · 2024-11-27

We appreciate the feedback provided by the reviewers and the Area Chair, and we are grateful for the opportunity to clarify key aspects of our work. Unfortunately, we haven't received any reviewer's response, thus we are writing to summarize a major clarification and our paper revision.

To start with, we appreciate all the reviewers’ recognition of our work's novelty in new problem formulation and solution. We believe there may be some misunderstandings regarding the scope and novelty of our proposed framework from some of the reviewers, particularly concerning the assumptions of data distribution drift. To clarify, our paper introduces Continual Multitask Learning (CMTL) as a new problem category focusing on label space iteration, where tasks arrive sequentially with distinct labels applied to a consistent input distribution. The misunderstanding appears to stem from the assumption that CMTL must address data distribution drift. However, the proposed CMTL setting deliberately focuses on a scenario where tasks are introduced via label space iteration over a fixed input distribution. Unlike traditional continual learning, which often assumes distribution drift, our framework specifically addresses challenges like catastrophic forgetting and task interference within the label space. This is a realistic assumption in applications like medical imaging or autonomous systems, where the data distribution remains stable while new iterations of data or annotations are incrementally added. Handling data distribution drift falls outside the scope of this work but could be a promising direction for future research.

We emphasize that the reviewer's concern about data distribution drift does not apply to our work, as the proposed setting intentionally focuses on tasks with a fixed input distribution. This simplification enables us to tackle unique challenges in CMTL, such as preserving shared representations across sequential label spaces. Moreover, most reviewers have recognized the novelty of our approach, including the dynamically weighted preservation loss to effectively retains knowledge. The following components are incorporated into the revised manuscript:

 - We have revised the introduction to better articulate the CMTL setting and its practical implications to avoid potential misunderstandings.

 - We have revised the caption and description of Figure 2 to provide a clear explanation of its components.

- The theoretical perspective on learning problems and the ablation study have been integrated into the main body.

- We have also fixed typos and minor errors that were present in the paper.


Also, we have included following additional experiments in the manuscript:
- We have incorporated recent and relevant continual learning methods into our comparative analysis.
   - Dual View Consistency (DVC) [Gu et al., 2022]
   - Online Bias Correction (OBC) [Chrysakis and Moens, 2023]

- We have conducted experiments using larger models and higher-resolution datasets (see ***Appendix D.6***).

- We have expanded our experiments to assess the applicability of LwP in different task sequence scenarios. The model was tested on scenarios where it continuously learns additional tasks after initially learning five tasks in a multitask learning manner (see ***Appendix D.7***)




**References**

Chrysakis, A., & Moens, M.-F. (2023). Online bias correction for task-free continual learning. *International Conference on Learning Representations (ICLR)*.


Gu, Y., Wang, Y., Wu, Z., Herrmann, C., & Herrmann, J. M. (2022). Not just selection, but exploration: Online class-incremental continual learning via dual view consistency. *Proceedings of the IEEE/CVF Conference on Computer Vision and Pattern Recognition*.

---

### Author Response · Authors · 2024-12-02

Dear Reviewers,

Thank you for reviewing our submission and providing constructive feedback. We have carefully addressed your comments in the rebuttal and believe these revisions help clarify the contributions and resolve potential misunderstandings. If our responses address your concerns, we kindly ask you to reconsider your evaluation. Should there be any remaining questions, we are happy to engage in further discussion and provide clarification before the Dec. 3rd deadline.

Best regards,

Author(s)

---

### Meta-Review · Area_Chair_K7gh · 2024-12-20

**Metareview:**

This paper received mixed reviews. The reviewers recognized the well designed method for the proposed continual multi-task learning setting, its strong performance in the setting, and extensive experiments. However, they disagreed on the value of the proposed problem setting and benchmarks: Two reviewers appreciated the value of the problem setting (minZ, KdBT), while the others argued that the problem setting as a variation of existing ones and thus pointed out incremental novelty (qi2x, 9JHi). Moreover, three reviewers considered that the benchmarks do not well reflect real-world application scenarios due to due to the lack of distribution shift, small-scale datasets, and outdated model architectures (qi2x, 9JHi, KdBT). Beside these issues, the reviewers raised concerns with lack of comparisons with latest work (qi2x, minZ), potential sensitivity to hyperparameters (qi2x), lack of theoretical foundation of the proposed loss function (minZ), and missing essential ablation study on the dynamic weighting (9JHi).

The authors' rebuttal and subsequent responses in the discussion period address some of these concerns but failed to fully assuage all of them. In particular, after the discussion period, Reviewer qi2x still pointed out the issues on the problem settings and lack of comparisons with latest work. Also, the AC found that the concerns with limitations of the benchmarks have been only partially resolved as the authors did not present additional datasets or incorporate the reviewers' suggestion, although Reviewer 9JHi did not come back and Reviewer KdBT gave a positive score in the end. Further, the AC sees that the rebuttal did not successfully address some of the remaining concerns like sensitivity to hyperparameters (quantitative analysis required) and the lack of theoretical foundation (only empirical analysis results added).

Putting these together, the AC considers that the remaining concerns outweigh the positive comments and the rebuttal, and thus regrets to recommend rejection. The authors are encouraged to revise the paper with the comments by the reviewers and the AC, and submit to an upcoming conference.

**Additional Comments On Reviewer Discussion:**

The rebuttal failed to assuage major concerns of the reviewers, and thus two reviewers voted to reject; one of the negative reviewers did not come back but the AC found that his/her concerns are not fully resolved by the rebuttal and revision. Below the major concerns of the reviewers and how they are addressed are summarized.

- **The proposed problem setting and benchmarks do not well reflect real world application scenarios (qi2x, 9JHi, KdBT)**: *The AC weighed this issue very heavily when making the final decision.* Reviewer qi2x considered the proposed problem setting, i.e., continual multi-task learning (CMTL), as a simplified variant of existing continual learning settings due to the absence of distribution shift, and thus believes the setting does not reflect real-world application scenarios; Reviewer KdBT also left almost the same comment, the reviewer was positive though. Reviewer 9JHi pointed out that the proposed benchmarks are limited due to the use of small-scale datasets and outdated model architectures. The authors failed to fully assuage these concerns. They did not provide additional experiments with distribution shifts, but only reiterated the significance of the problem setting mentioned already in the paper. Also, additional experiments in the rebuttal failed to address the concerns with the benchmarks, in particular their scales and reality, since the experiments was conducted on one of the datasets already used in the paper. For these reasons, the value of the CMTL setting does not look significant to this AC. In particular, it is unclear what is its key difference from class incremental learning as the benchmarks are still about classification, i.e., classification of different types of attributes, which however seems to be interpreted as a variant of class incremental learning. It would be nice if the authors demonstrate more task variations to reflect more realistic application scenarios illustrated in the rebuttal and revision.
- **Lack of comparisons with latest work (qi2x, minZ)**: The AC feels this issue has been well addressed by additional results in the rebuttal, and Reviewer minZ was also satisfied. However, Reviewer qi2x did not; the reviewer wanted comparisons with additional state-of-the-art methods. *The AC did not weigh this issue heavily when making the final decision* since the last comment by Reviewer qi2x does not suggest any specific example of such methods to be compared.
- **Potential sensitivity to hyperparameters (qi2x)**: The reviewer said his/her concern on this issue has been resolved, but the AC does not agree. Using the same set of hyperparameter values for all the three datasets is of course desirable, but still it could be difficult to find such a combination of hyperparameter values that works best for all the three datasets, especially if the performance is sensitive to the hyperparameters. To better address this issue, the authors should provide detailed quantitative analysis results, e.g., accuracy vs. hyperparametere values.
- **Lack of theoretical foundation of the proposed loss (minZ)**: The AC sees this issue has not been fully addressed since the authors did not provide theoretical foundation but instead presented ablation study on the loss function. However, the reviewer seems to be satisfied as he/she raised the score to 6, and thus *the AC did not weigh this issue heavily when making the final decision*
- **Missing essential ablation study on the dynamic weighting (9JHi)**: Additional results in the rebuttal and revision successfully resolved this issue.

**Misc**: The AC found that the most negative reviewer is the most confident and the most experienced in the continual learning field among the four reviewers. Also, the quality of writing should be improved substantially to meet the standard of ICLR and other top-tier ML conferences.

---

### Decision · Program_Chairs · 2025-01-22

Reject